# Fitness drivers of division of labor in vertebrates

**Irene García-Ruiz\*, Dustin R Rubenstein**

Department of Ecology, Evolution and Environmental Biology, Columbia University, New York, United States

## eLife Assessment

This **useful** study develops an individual-based model to investigate the evolution of division of labor in vertebrates, comparing the contributions of group augmentation and kin selection. The findings are **solid** in showing that, within the specific structure of the model and the parameter space explored, group augmentation can robustly favor the evolution of differentiated helper roles, particularly when age-dependent task switching and dominance dynamics are allowed to evolve. However, the evidence only partially supports the authors' broader claim that group augmentation is the primary driver of vertebrate division of labor. Several modelling assumptions, including the limited scope for synergistic task benefits, the restriction of helper effects to group-size-mediated benefits, and the relatively narrow exploration of cost and benefit parameters, constrain the potential for kin selection to generate division of labor and limit the generality of the conclusions.

**\*For correspondence:**
igaru.13@gmail.com

**Competing interest:** The authors declare that no competing interests exist.

**Abstract** Although division of labor as a means to increase productivity is a common feature in animal social groups, most previous studies have focused almost exclusively on eusocial insects with extreme task partitioning. Empirical evidence of division of labor in vertebrates is scarce, largely because we lack a theoretical framework to explore the conditions under which division of labor is likely to evolve in cooperatively breeding systems where helpers remain capable of breeding throughout their lifetime. By explicitly considering alternative helping tasks with varying fitness costs, we model how individual decisions on task specialization may influence the emergence of division of labor under both direct and indirect fitness benefits. Surprisingly, we find that direct survival benefits of living in larger groups are the primary force driving the evolution of cooperation to enhance group productivity, and that indirect fitness benefits derived from related group members are only a non-essential facilitator of more stable forms of division of labor in cooperative breeders. In addition, we find that division of labor in vertebrates is favored by harsh environments. Ultimately, our model not only makes key predictions that are consistent with existing empirical data, but also proposes novel avenues for new empirical work in vertebrate and invertebrate systems alike.

## Introduction

Division of labor as a means to boost productivity holds significance not only in human societies, but also those of other animals (*Smith, 1776*; *Oster and Wilson, 1978*; *Dunbar et al., 2014*; *Zhang et al., 2016*; *Cooper and West, 2018*). Nearly 250 years ago, the economist *Smith, 1776*, proposed that division of labor via task specialization within human societies not only improves individual efficiency and group productivity, but also reduces the costs associated with switching tasks. Similarly, in nonhuman animal societies, division of labor to increase group reproductive output (analogous to economic gain) is characterized by group members specializing in particular tasks, either temporarily or permanently (*Michener, 1974*; *Beshers and Fewell, 2001*; *Jeanne, 2016*;). To date, most

theoretical and empirical work on the selective pressures favoring division of labor via task specialization in animal societies has focused on 'classically' eusocial insect societies characterized by permanently sterile workers that are never able to reproduce (*Wheeler, 1928*; *Wilson, 1971*; *Beshers and Fewell, 2001*; *Willensdorfer, 2009*; *Ispolatov et al., 2012*; *Li et al., 2025*). Since workers in eusocial societies can, in most cases, only obtain fitness benefits indirectly through kin selection by maximizing their colony reproductive output, selection can occur at the colony level, typically resulting in very low conflict of interest between group members. However, complex societies occur outside of eusocial insects, such as in many birds and mammals, including humans. Because sterile workers are absent in vertebrates and other cooperative breeding societies, conflict can be much higher than in most eusocial insect societies. The few studies of division of labor outside of eusocial insects have primarily focused on reproductive division of labor (i.e. reproductive skew), despite that fact that temporal or permanent non-reproductive organisms can engage in an array of different helping tasks (*Cooper and West, 2018*; *Yanni et al., 2020*; *Taborsky et al., 2025*). Given the scarcity of studies examining the selective advantages of division of labor beyond eusocial insects (but see *Ridley and Raihani, 2008*), our understanding of the evolutionary pressures of division of labor is largely confined to instances where task specialization is linked to sterility and where conflict of interest is generally low (*Wenseleers et al., 2021*).

In cooperatively breeding vertebrates, which typically live in societies with more equitable sharing of reproduction than eusocial insects (i.e. low reproductive skew), low or mixed kinship, and often have unrelated helpers, conflict of interest can be high, and direct fitness benefits may be more relevant in the evolution of division of labor than indirect benefits (*Dunn et al., 1995*; *Doutrelant et al., 2011*; *Riehl, 2013*). In fact, shared fitness incentives to increase group productivity are not limited to eusocial insects with high within-group relatedness and sterile workers. In cases where group size positively correlates with an increase in the survival or reproduction of group members, division of labor via task specialization may also evolve under direct fitness benefits to subordinate members of a group (i.e. group augmentation; *Kokko et al., 2001*; *Kingma et al., 2014*). Although a great deal of recent empirical work highlights the importance of direct benefits in the evolution of cooperative breeding behavior in vertebrates (*Richardson et al., 2002*; *Sparkman et al., 2011*; *Zöttl et al., 2013b*; *Shah and Rubenstein, 2023*), we still lack an understanding of the joint influence of direct and indirect fitness benefits in the evolution of division of labor.

Understanding how direct and indirect benefits interact is particularly important in systems where individuals may differentially bear the fitness costs of cooperation. Most previous models of division of labor via task specialization considered all helping tasks under the same fitness umbrella, which is a good approximation in eusocial insect societies because colony members share fitness costs and benefits. In vertebrates and social insects with fertile workers, however, helpers may obtain fitness benefits directly via reproduction, and thus, selection occurs more strongly at the individual level than it does in most eusocial insects. Group members in vertebrate societies may then contribute to distinct helping tasks differently according to individual life-history strategies (*Cockburn, 1998*; *Anderson, 2001*; *Beehner and Kitchen, 2007*) or differences in the magnitude or type of fitness costs and benefits that they receive from cooperative tasks (*Nunn and Lewis, 2001*). One of the key distinctions between fitness costs is whether a task impacts more immediate survival or direct reproduction. For instance, defensive behaviors such as predator mobbing may have a high risk of injury and death (*Poiani and Yorke, 1989*; *Sordahl, 1990*; *Tórrez et al., 2012*), whereas other activities such as allofeeding or nest/territory maintenance may affect body condition owing to lower feeding rates and time and energy investment, consequently affecting individual reproductive output (*Heinsohn and Cockburn, 1994*; *Grantner and Taborsky, 1998*; *Heinsohn and Legge, 1999*). Therefore, changes in life histories, such as those linked to the likelihood of becoming a breeder, may select for task specialization according to the relative direct fitness cost associated with different helping tasks.

In addition to life-history differences, demographic, social, and ecological factors that affect helper workload and the prospect of helpers becoming breeders are likely to have a strong impact on task allocation because they alter the probability of obtaining direct fitness benefits (*Young et al., 2005*; *Zöttl et al., 2013a*). For example, age is known to be an important factor influencing helping workload in vertebrates (*Komdeur, 1996*; *Boland et al., 1997*), is often correlated with differences in other life-history traits (*Heinsohn and Cockburn, 1994*; *Taborsky, 1984*; *Boesch, 2002*; *Clutton-Brock et al., 2003*), and may be a proxy for other processes involved in the evolution of help (*Johnstone*

*and Cant, 2010*; *Fischer et al., 2014*). Moreover, age-dependent task allocation (i.e. temporal polyethism) is common in eusocial insects that do not show morphological differentiation (*Hölldobler and Wilson, 1990*). Social and ecological factors are also thought to influence the emergence of division of labor in eusocial insects (*Sendova-Franks and Franks, 1993*; *Traniello and Rosengaus, 1997*; *Ferguson-Gow et al., 2014*). Indeed, high relatedness may reduce conflict of interest between group members and promote the evolution of division of labor (*Queller and Strassmann, 1998*; *Gardner and Grafen, 2009*; *Keller and Chapuisat, 2010*; *Fisher et al., 2013*; *West et al., 2015*). Similarly, in harsh environments where group size strongly influences survival (*Rood, 1990*; *Clutton-Brock et al., 1999*; *Taborsky et al., 2005*; *García-Ruiz and Taborsky, 2022*), division of labor may be favored to more efficiently increase group productivity.

To disentangle the role of direct versus indirect fitness benefits in the evolution of division of labor in cooperative breeding societies with totipotent helpers, we created an individual-based model in which we varied the influence of different evolutionary forces to understand social and ecological circumstances under which division of labor may arise to maximize the inclusive fitness of individuals in a group. We used distinct tasks with different fitness costs to maximize the group reproductive output when they were performed to a similar extent. Individuals in our model were allowed to evolve different behavioral responses or reaction norms in task specialization, depending on their probability of becoming breeders. We compared the results of this model to those of a benchmark model in which the reproductive outcome was independent of the type of task performed by group members. We present the results for different environmental conditions in the presence or absence of relatedness (indirect) or group size (direct) benefits. Ultimately, this work demonstrates that the direct fitness benefits of group living attained in increasingly harsh environments are key to the evolution of division of labor in vertebrates.

## Results

### Rationale

Our model consists of a population characterized by overlapping generations residing in a habitat featuring a fixed finite number of breeding territories. These territories are monopolized by groups composed of a single breeder and a variable number of subordinate helpers, the number of which is shaped by population dynamics, with all group members capable of reproducing during their lifetime. Individuals in the population fall into one of three categories: (1) *breeders* that monopolize reproduction within a group; (2) *subordinates* that may perform different helping tasks, either in their natal group or in an outside group to which they dispersed; and (3) *floaters* that disperse from their natal group but remain not part of a group.

We assume complete reproductive skew such that subordinate group members are prevented from reproducing at a given timestep and may only help raise the offspring of the dominant breeders. Dispersers cannot reproduce without acquiring a territory (denoted here as floaters). Helpers are not sterile and may inherit the breeding position when the breeder dies. When this occurs, helpers and dispersers from other groups compete for the breeding position and win with a probability proportional to their dominance value (*Figure 1*; *Table 1*). We use the term dominance value to designate the competitiveness of an individual compared to other candidates in becoming a breeder, regardless of group membership. Dominance value increases as a function of age, serving as a proxy for resource holding potential (RHP), and decreases as a function of help provided, reflecting costs to body condition from performing working tasks (*Equation 2*).

Subordinate helpers in our model may engage in one of two distinct tasks with differing fitness costs: (1) *defensive tasks* (i.e. defense of the dominant breeders' offspring at cost to their own survival); or (2) *work tasks* (i.e. tasks with an associated cost to their body condition and consequently to their competitiveness to become breeders in the future). Subordinate helpers can then evolve different workloads and exhibit task specialization that is either fixed throughout their lifetime or varies with their dominance value. The parameters $x_h$ and $y_h$ (*Table 1*) reflect how much the subordinates' survival and dominance values decrease with increasing investment in defense or work tasks, respectively. Although we do not assume an inherent difference in body condition between individuals when born, variance in age and helping propensities can create differences in the fitness costs linked to task performance.

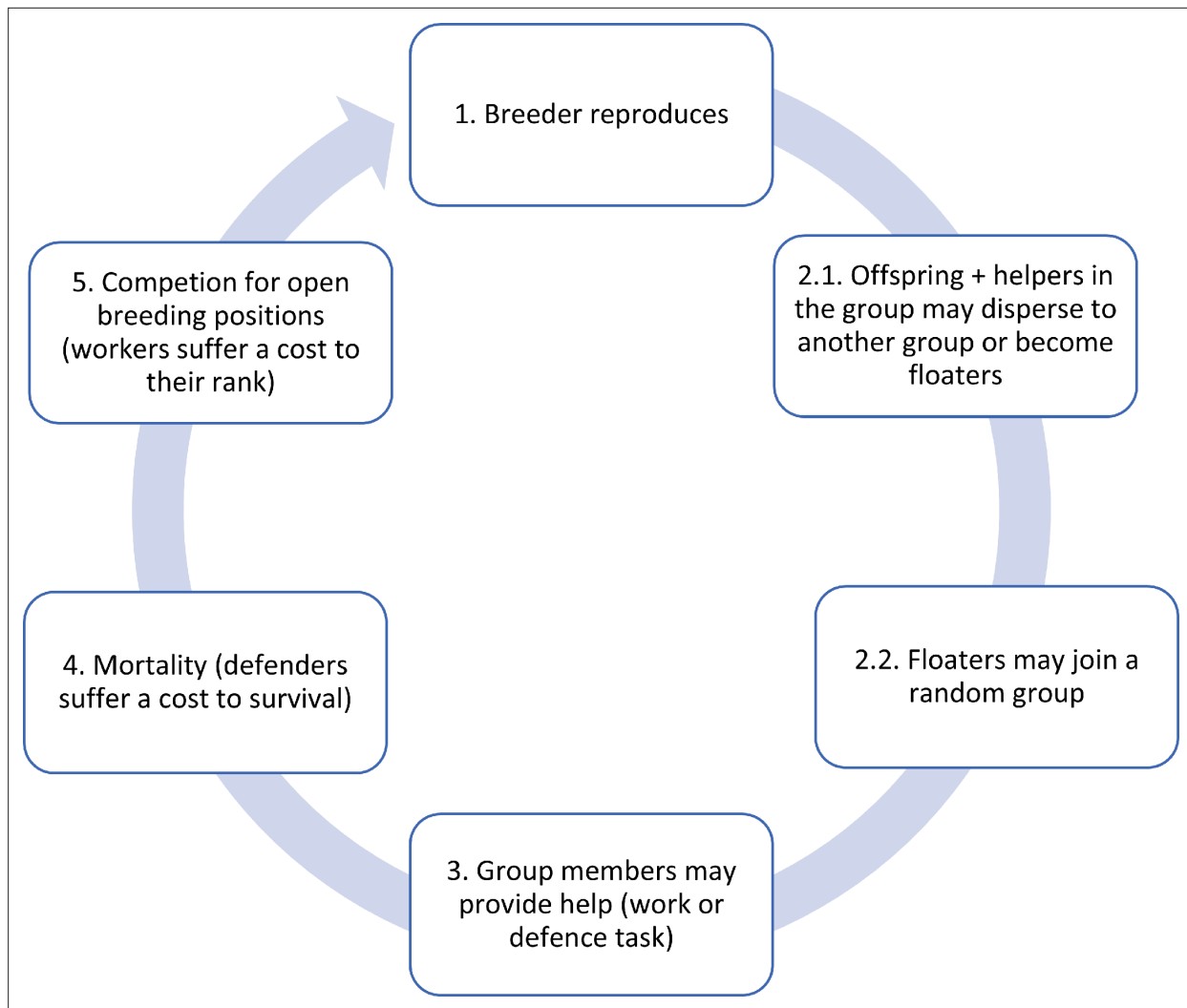

**Figure 1.** Diagram of the scheduling executed per breeding cycle. (1) A breeder reproduces. Its productivity depends on the cumulative level of brood care provided by the group during the previous breeding cycle. Maximum productivity is achieved when different helping tasks are performed to a similar extent. (2) Subordinates may disperse to become floaters, or they may stay in the group and help (1). Dispersers/floaters may join a random group to become subordinates. (3) Subordinates in the group (both natal and immigrant individuals) either work to provision to the breeder's offspring or display defensive forms of help. (4) Individuals survive contingent on group-living benefits and dispersal costs, as well as the cost of defensive activities. (5) If a breeder dies, helpers in the group and a sample of floaters compete for the breeding position. Individuals still alive ascend one age class, and the cycle starts all over (i.e. next breeding cycle).

We assume net survival benefits from living in large groups (i.e. group augmentation), which occurs in many social species (*Rood, 1990*; *Taborsky et al., 2005*; *Bilde et al., 2007*; *Guindre-Parker and Rubenstein, 2020*). Subordinate members of a group may therefore gain both direct and indirect fitness benefits from helping to increase the productivity of breeders, if doing so results in an increase in group size. In the model, the benefit of alloparental care to the productivity of the breeder is greatest when group members provide overall similar workloads for both defensive and work tasks, since both are often needed for the survival of the offspring (*Rubenstein et al., 2016*; *Taborsky, 2016*). For example, in many cooperatively breeding birds, the primary reasons that breeders fail to produce offspring are (1) starvation, which is mitigated by the feeding of offspring, here considered as a work task, and (2) nest depredation, which is countered by defensive behavior (*Hatchwell, 1999*; *Rubenstein et al., 2016*). Consequently, both types of tasks are often necessary for successful reproduction, and focusing solely on one while neglecting the other is likely to result in lower reproductive success than if both tasks are performed by helpers within the group. To compare between the helper's preferred task and the task needed to maximize group productivity, we created a benchmark model in

**Table 1.** Overview of notation.

Values conveyed for the genes are initial input values, and values given for the scaling parameters are fixed throughout the simulations. If more than one value is given, results are shown to display the effect of this parameter's variation.

| Symbol | Meaning | Value |
|---|---|---|
| *Genes* | | |
| $\alpha$ | Genetic propensity to help | 0 |
| $\beta$ | Genetic predisposition to disperse vs remain in a group | 1 |
| $\beta_0$ | Intercept in the dispersal reaction norm | 1 |
| $\beta_R$ | Effect size of rank on dispersal propensity | 0 |
| $\gamma_0$ | Intercept in the task specialization reaction norm | 0 |
| $\gamma_R$ | Effect size of rank on task specialization | 0 |
| | | |
| *Scaling parameters* | | |
| $y_h$ | Effect size of cost of help on dominance when performing 'work tasks' | 0.1* |
| $x_h$ | Effect size of cost of help in survival when performing 'defensive tasks' | 3 / 5 / 7* |
| $x_n$ | Effect size of the benefit of group size in survival | 0 / 3 |
| $x_0$ | Intercept in the survival function | 1.5* |
| $m$ | Baseline mortality | 0.1 – 0.3 |
| $f$ | Mean number of groups a floater samples for becoming a breeder | 2* |
| $k_m$ | Deviation from a perfect split in the need of both defense and work tasks | 0.1 |
| $k_h$ | Effect size of the cumulative help of subordinates on the fecundity of the breeder | 1* |
| $k_0$ | Fecundity of the breeder in the absence of help | 1 |
| $\mu$ | Mutation rate | 0.05 |
| $\sigma_\mu$ | Mutation step size | 0.04 |
| $N_b$ | Number of breeding territories | 5000 |
| | | |
| *Phenotypic traits* | | |
| $H$ | Level of help of either kind provided to the breeder $H = \alpha$, applying boundary of 0 for negative numbers. We express cumulative level of help given in a group as $\sum H_i$ | |
| $D$ | Dispersal propensity $D = \beta$, applying boundaries between 0 and 1 | |
| $R$ | Dominance value, applying boundary of 0 for negative numbers | |
| $T$ | Probability of choosing 'defensive task' vs 'work task' | |
| $S$ | Survival probability | |
| $K$ | Fecundity of the breeder | |
| | | |
| *Variables* | | |
| $t$ | Age | |
| $N$ | Emergent group size | |
| $N_f$ | Total number of floaters | |
| $N_{f,b}$ | Number of floaters bidding for a breeding position, given by $N_{f,b} = f * N_f/N_b$ | |

*Additional values of these parameters are discussed in the Appendices. A broader parameter landscape was explored but not included in the manuscript.

which the breeder's productivity is not constrained by the type of help provided. Higher productivity translates to larger group sizes, which enhances the survival prospects of all group members. Hence, subordinates have an incentive to develop division of labor. Subordinates that remain in their natal group may also gain improved fitness by indirectly passing on their genes through kin. Both group size and relatedness in our model are emergent properties of a series of demographic processes, including mortality, dispersal, and help-dependent breeder fertility.

To distinguish between the effects of group augmentation and kin selection on inclusive fitness, we created two additional models. In the first model, we removed the benefits of group size (group augmentation) by setting the parameter $x_n = 0$ (*Table 1*). In the second model, we removed the effect of kin selection by simulating a cross-foster manipulation in which offspring just born are interchanged between different groups (see Materials and methods for additional details). To assess how habitat quality affects task specialization and the emergence of division of labor, we systematically varied habitat quality across simulations. Harsh environments are those that have high mortality (i.e. $m = 0.3$; *Table 1*), which also raises the turnover of the breeding positions. Lower values of $m$ reflect an increased probability of individuals surviving another breeding cycle and, therefore, the chances of habitat saturation. The steps in the breeding cycle of the model are illustrated in *Figure 1*.

## Direct versus indirect fitness benefits

We first examined how both direct fitness benefits from group living and indirect fitness benefits from aiding kin influence the evolution of division of labor through task specialization across environments of varying quality. Our results suggest that voluntary division of labor involving tasks with different fitness costs is more likely to emerge initially because of direct fitness benefits (*Figure 2*; GA), but a combination of direct and indirect fitness benefits leads to higher rates and more stable forms of division of labor (*Figure 2*; KS+GA). It is important to note that, as depicted in *Figure 2*, intermediate values of task specialization indicate in all cases age/dominance-mediated task specialization ($\gamma_R \neq 0$; *Table 1*) and never a lack of specialization ($\gamma_R = 0$; *Table 1*); further details are shown in the next section: '*The role of dominance in task specialization*'.

In contrast to expectations from models of eusocial insects, our vertebrate model does not readily evolve division of labor when only kin selection is considered (see Appendix 2 for more details on further exploration on conditions that may favor the evolution of division of labor under only kin selection). Specifically, forms of help that impact survival rarely evolve under any environmental condition when only kin selection occurs (*Figure 2*; KS). This occurs because with kin competition for the breeding position and no survival benefits of group membership, the reduced incentives to remain in the natal group drive most subordinates to disperse and breed elsewhere (average dispersal rate between 88% in harsh environments and 93% in benign environments; *Table 2*). Adding a reaction norm of dispersal to dominance rank did not change the results (see Appendix 3). The few subordinates that remain in the natal territory help to obtain indirect benefits (average within-group relatedness between 0.36 in benign environments and 0.72 in harsh environments; *Table 2*). Even though individuals in our model cannot adjust their help according to their relatedness to the breeder, the high levels of relatedness that evolved under this paradigm indicate that allowing for targeted help would not qualitatively change the results. Survival costs of defensive tasks are then avoided, as mortality without group benefits is relatively high, while costs to dominance value are small for close relatives because of indirect fitness benefits and low conflict of interest (*Figure 2*; KS). Adjusting input parameters to reduce defense costs on survival or to increase survival probabilities and favor philopatry did not qualitatively change the results, except for extreme values of baseline survival (*Appendix 2—figures 1 and 2*). Only when (1) there is a tenfold increase in the productivity of the breeders per unit of help, and (2) division of labor is required for a productivity boost, do group members stay philopatric and coordinate labor to help raise kin in some circumstances (*Appendix 2—figure 4*). Hence, kin selection alone is unlikely to select for the evolution of defensive tasks and division of labor in vertebrates.

Conversely, under only direct fitness benefits (group augmentation), tasks associated with costs to the dominance value are costlier because the only path to obtain fitness benefits is via direct reproduction. Therefore, defensive tasks that do not affect the dominance value are favored (*Figure 2* and *Appendix 4—figure 1*; GA). Under this scenario, division of labor can evolve to maximize the reproductive output of the breeders (*Figure 2*; GA, $m = 0.3$, $x_h \leq 5$), but also because it is the optimal strategy (*Figure 2*; GA, $m = 0.3$, $x_h = 7$). In benign environments, however, no help evolves due to

the lower incentives to increase survival and the steep competition for reproduction resulting from habitat saturation.

When subordinates can obtain both direct and indirect fitness, division of labor is more strongly selected in increasingly harsher environments to increase breeder productivity, a result mainly driven by an increase in direct fitness benefits (*Figure 2*; KS + GA). Conversely, when direct fitness benefits are low, division of labor primarily emerges in benign environments from a combination of both direct and indirect fitness benefits (*Appendix 4—figure 1*; KS + GA). Additionally, in moderate and harsh environments, work tasks with costs to the dominance value are more strongly selected than in benign environments due to high mortality increasing the breeder's turnover, which reduces breeding competition (lower effective costs of work tasks), and the costs to survival (high costs of defensive tasks), while defense tasks are mainly promoted by direct fitness benefits and in more benign conditions (*Appendix 4—figure 1*). Hence, in more benign (and often highly productive) environments that lead to habitat saturation, helping likely evolved initially in family groups, and defensive tasks are favored

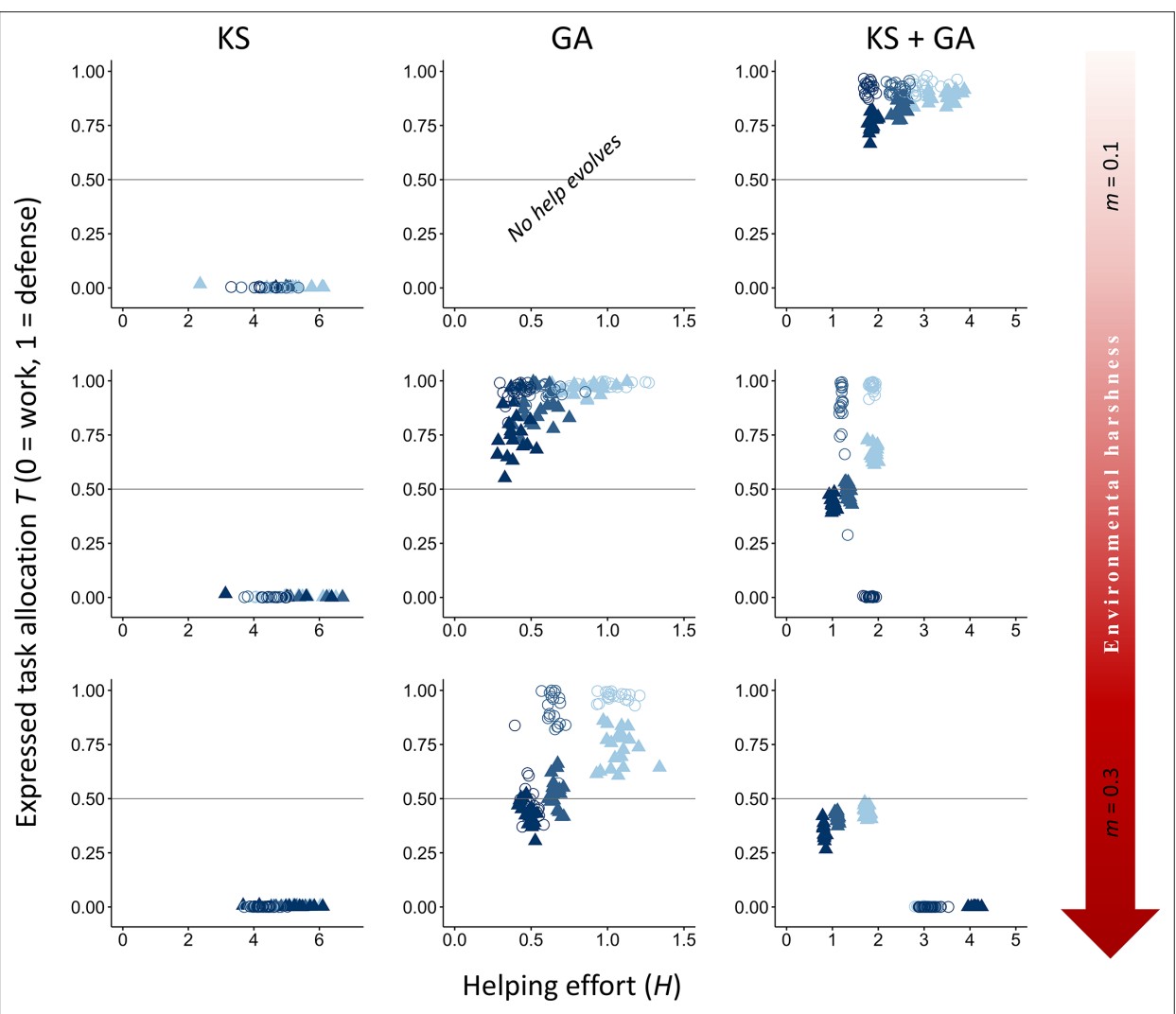

**Figure 2.** Effect of environmental quality on alloparental care and division of labor. The evolutionary equilibria for phenotypic levels of helping and task specialization are shown at three different levels of environmental quality, ranging from benign ($m = 0.1$) to harsh ($m = 0.3$), and for three different levels of cost of help on survival (light blue, $x_h = 3$; blue, $x_h = 5$; and dark blue, $x_h = 7$), across 20 replicas. The vertical axis represents the expressed task allocation between a defensive task with a cost to survival versus a work task with a cost to their dominance rank. The optimum breeder productivity per unit of help provided was either when both tasks were performed to a similar extent, potentially selecting for division of labor (▲) or when no restrictions were introduced to the task performed by the group members (○). In each environment, additional details are given on the selective forces that play a role in the evolution of help and task specialization: help can only evolve by kin selection (KS), group augmentation (GA), or both (KS + GA). Additional details are provided in *Table 2*. For variation in $y_h$ values instead, see Appendix 1. All input parameter values are described in *Table 1*.

**Table 2.** Supplementary data for *Figure 2*.

Mean values are shown for dispersal propensity, survival probability, group size (± SD), number of floaters (± SD), ratio between helpers' and floaters' dominance value (helpers' rank/floaters' rank ± SD) and within-group relatedness for three environmental qualities ranging from benign ($m = 0.1$) to harsh ($m = 0.3$) across 20 replicas. Selective forces at play include kin selection (KS), group augmentation (GA), or both (KS + GA). The optimum breeder productivity per unit of help provided was either when both tasks were performed to a similar extent, potentially selecting for division of labor (DoL), or when no restrictions were introduced to the task performed by the group members (No DoL).

| | KS | | | | | | GA | | | | | | KS +GA | | | | | |
|---|---|---|---|---|---|---|---|---|---|---|---|---|---|---|---|---|---|---|
| | m = 0.1 | | m = 0.2 | | m = 0.3 | | m = 0.1 | | m = 0.2 | | m = 0.3 | | m = 0.1 | | m = 0.2 | | m = 0.3 | |
| | DoL | No DoL | DoL | No DoL | DoL | No DoL | DoL | No DoL | DoL | No DoL | DoL | No DoL | DoL | No DoL | DoL | No DoL | DoL | No DoL |
| Dispersal | 0.95 | 0.92 | 0.94 | 0.90 | 0.91 | 0.84 | 0.01 | 0.01 | 0.11 | 0.10 | 0.37 | 0.33 | 0.07 | 0.07 | 0.22 | 0.22 | 0.43 | 0.52 |
| Survival | 0.74 | 0.74 | 0.65 | 0.65 | 0.57 | 0.57 | 0.90 | 0.90 | 0.79 | 0.79 | 0.66 | 0.67 | 0.89 | 0.89 | 0.77 | 0.77 | 0.65 | 0.65 |
| Group size | 1.16±0.16 | 1.34±0.30 | 1.14±0.12 | 1.25±0.19 | 1.14±0.08 | 1.29±0.11 | 9.85±0.08 | 9.83±0.05 | 6.94±0.43 | 7.52±0.41 | 3.04±0.35 | 3.40±0.36 | 16.42±0.10 | 16.55±0.07 | 6.42±0.14 | 6.61±0.11 | 2.95±0.25 | 2.83±0.05 |
| Number of floaters | 14280±1352 | 15616±2322 | 9778±711 | 10352±1082 | 6956±330 | 7482±395 | 297±23 | 297±22 | 3512±104 | 3428±101 | 5925±137 | 5905±122 | 5956±121 | 5968±107 | 7523±252 | 7853±201 | 7353±1173 | 9741±224 |
| Rank ratio helpers vs floaters | 0.91±0.27 | 0.94±0.22 | 0.85±0.24 | 0.88±0.25 | 0.76±0.27 | 0.75±0.12 | 1.05±0.06 | 1.06±0.06 | 1.05±0.04 | 1.06±0.04 | 1.03±0.06 | 1.03±0.06 | 1.02±0.03 | 1.03±0.03 | 1.00±0.04 | 1.00±0.04 | 0.97±0.07 | 0.89±0.06 |
| Relatedness | 0.37 | 0.36 | 0.54 | 0.52 | 0.73 | 0.71 | 0 | 0 | 0 | 0 | 0 | 0 | 0.56 | 0.57 | 0.57 | 0.57 | 0.65 | 0.60 |

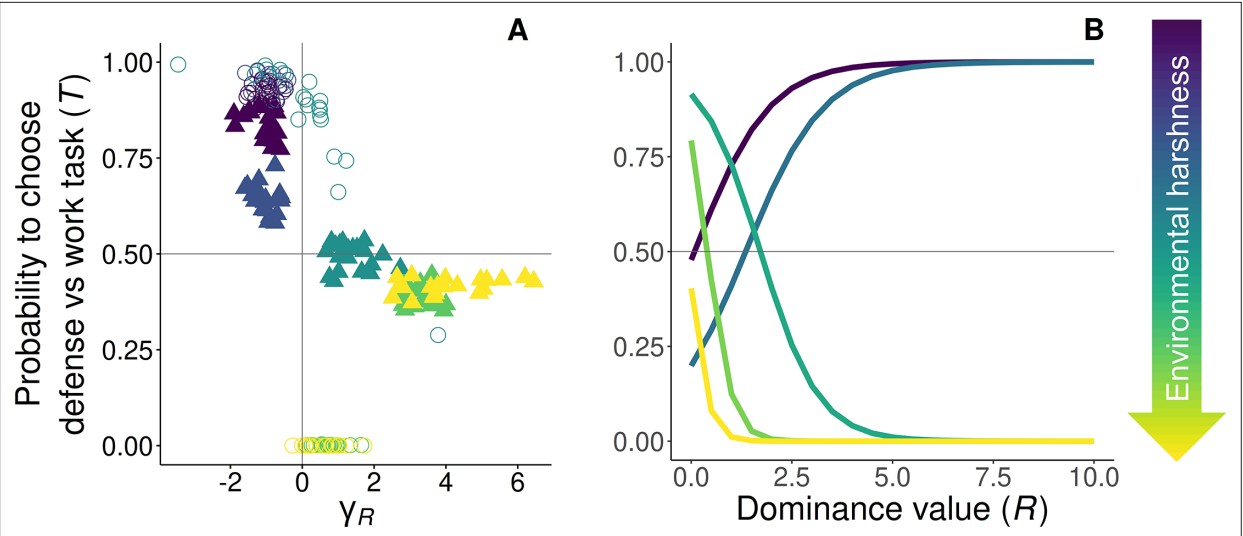

**Figure 3.** Evolved reaction norms to age on the display of task specialization. The evolutionary equilibria for the reaction norms of task specialization are shown at five different levels of environmental quality, ranging from benign ($m = 0.1$, purple) to harsh ($m = 0.3$, yellow), and $x_h = 5$. (**A**) $\gamma_R > 0$ signifies that individuals increase the probability of performing work tasks with dominance (defense → work), whereas $\gamma_R < 0$ signifies that individuals increase the probability of performing defensive tasks with dominance (work → defense). The optimum breeder productivity per unit of help provided was either when both tasks were performed to a similar extent, potentially selecting for division of labor (▲), or when no restrictions were introduced to the task performed by the group members (○). Results show that at equilibrium when division of labor evolves, individuals increase the probability of performing their preferred task (circles), when increasing their dominance value. (**B**) Evolved reaction norms to dominance value with average $\gamma_0$ and $\gamma_R$ across 20 replicas for varying quality environments. All parameter values described in **Table 1**.

because competition for the breeding position is lower under kin selection. In this scenario, kin selection acts not only by favoring subordinates in their natal group to increase the productivity of a related breeder (**Figure 2**; KS), but also by the increased survival of their siblings when augmenting the group size (**Figure 2**; KS + GA), regardless of whether they are related to the current breeder. In harsher environments, however, competition between unrelated individuals decreases in favor of coordinated cooperation to reduce mortality (**Figure 2** and **Appendix 4—figure 1**; GA). Together, these patterns indicate that direct fitness benefits are the primary force driving cooperative specialization in harsh environments, where increased breeder productivity boosts the survival of both helpers and their relatives (**Figure 2**). In contrast, in benign environments, cooperation arises mainly through kin-selected benefits, with subordinates contributing primarily by performing their preferred task of defending the breeder's offspring (**Figure 2**).

## The role of dominance in task specialization

In the previous section, we explored the role of direct and indirect fitness benefits in the evolution of division of labor via task specialization. However, determining whether parameter $\gamma_R$ (i.e. the direction and rate of change in task specialization with dominance) significantly differs from zero is crucial. Intermediate values of phenotypic task specialization may indicate division of labor among different age- or dominant-related groups (if $\gamma_R \neq 0$), or lack of specialization in either task (if $\gamma_R = 0$). Hence, for age-polyethism or temporal division of labor to arise in our model, individuals must switch between tasks based on their age-related dominance (probability of choosing a 'defensive task' versus a 'work task' $T \neq 0$ or $1$, and $\gamma_R \neq 0$; **Table 1**). Across all scenarios, individuals are selected to perform tasks based on their probability of inheriting the territory (**Figure 3A**). Specifically, when subordinates prefer defensive tasks (**Figure 3A**, $m < 0.2$, circles), they engage in work tasks at a young age and increasingly undertake defensive tasks as their dominance rises (**Figure 3**, $m < 0.2$). Similarly, if their preference leans toward work tasks (**Figure 3A**, $m \geq 0.2$, circles), younger subordinate helpers and those of lower dominance focus more on defense initially, but switch to work tasks as they approach the breeding stage (**Figure 3**, $m \geq 0.2$). In essence, subordinate helpers allocate more time to tasks with lower fitness costs as they near a competitive dominance value, all without any active enforcement by other group members, resulting in age- or rank-polyethism. These results are consistent with the rise

in division of labor, defined as within-individual consistency and between-individual differences in task choice (*Ulrich et al., 2018*).

## Discussion

Our findings suggest that the direct benefits of group living play a driving role in the evolution of division of labor via task specialization in species with totipotent workers, including all vertebrates and a few insects. In contrast, in eusocial species characterized by high relatedness and permanent worker sterility, such as in most eusocial insects, workers acquire fitness benefits only indirectly, and so, their fitness is contingent upon the overall success of the colony rather than the individual (*Foster et al., 2006*; *Hughes et al., 2008*). Group members in such eusocial species are therefore predicted to maximize colony fitness due to the associated lower within-group conflict and to favor the evolution of division of labor to increase group productivity efficiency (*Gardner and Grafen, 2009*; *Fisher et al., 2013*; *West et al., 2015*). However, in taxa with lower within-group relatedness, such as nearly all vertebrates, division of labor is facilitated when cooperation is crucial for survival and when the benefits of cooperation are evenly distributed among group members (*Cooper and West, 2018*). In particular, survival fitness benefits derived from living in larger groups seem to be key for the evolution of cooperative behavior in vertebrates (*García-Ruiz et al., 2022*; *Shah and Rubenstein, 2023*;), and may also translate into low within-group conflict. This suggests that selection for division of labor in vertebrates is stronger in smaller groups, where increased productivity significantly enhances survival, as opposed to research on eusocial insects that posits division of labor is more likely to evolve in larger groups (*Dornhaus et al., 2012*; *Nakahashi and Feldman, 2014*; *Ulrich et al., 2018*). That is not to say, however, that kin selection is unimportant in cooperatively breeding vertebrates or insects. Our model demonstrates that cooperative breeders are more likely to evolve division of labor when direct and indirect fitness benefits act in concert. Moreover, our model predicts that the type of task that individuals prefer depends on the type of fitness benefit that subordinates can attain. While direct fitness benefits in the form of group augmentation select more strongly for defensive tasks, kin selection alone seems to select only for work tasks. In groups of unrelated individuals where members can only gain direct fitness benefits, costs to dominance value and to the probability of attaining a breeding position are comparatively larger than survival costs. Conversely, groups with no group benefits tend to be small and suffer high mortality, for which defensive tasks only aggravate the situation.

Previous models of task allocation in eusocial insects have focused on how seemingly identical individuals of the same age and broad morphology may specialize in different types of tasks (*Ulrich et al., 2018*; *Beshers and Fewell, 2001*). In contrast, our model shows that extrinsic ecological factors are also important for the evolution of task specialization in addition to intrinsic factors such as age or dominance rank. In short, our model predicts that task specialization, which varies with age or rank, will depend on environmental quality. In more favorable environments, helpers are predicted to increase defensive tasks with age or rank, whereas in harsh environments, work tasks are predicted to increase with age or rank. Furthermore, harsher environments exert stronger selective pressure for task specialization due to increased survival challenges. Enhanced cooperation and improved group productivity through helping mitigate the negative effects of the environment. This finding is consistent with previous research on the role of ecological constraints in driving the evolution of cooperation (*Emlen, 1982*; *Hatchwell and Komdeur, 2000*; *Kokko and Ekman, 2002*).

Although rare, evidence of division of labor in cooperative breeding vertebrates comes from a few studies in mammals (*Lacey and Sherman, 1996*), birds (*Arnold et al., 2005*; *Ridley and Raihani, 2008*), and fishes (*Bruintjes and Taborsky, 2011*; *Josi et al., 2020*). Our model predictions are broadly consistent with empirical data from many of these studies. For example, support for the role of environmental harshness on the evolution of division of labor via age-dependent task specialization in vertebrates comes from arguably the most similar vertebrate system to eusocial insects, the naked mole-rat (*Heterocephalus glaber*). Naked mole-rats live in self-dug burrows that protect their inhabitants from predation and climatic extremes (*Lacey and Sherman, 1996*). Therefore, even though they inhabit habitats with scarce and unpredictable rainfall, environmental harshness is likely to be limited in their stable subterranean burrows, with mortality mainly spiking during dispersal attempts. Model predictions for species that evolved in benign environments are then met, with individuals shifting from work tasks such as foraging for food, digging, and maintaining the burrow system, to defensive tasks such as guarding and patrolling as individuals grow older and larger (*Faulkes et al.,*

*1991*; *Lacey and Sherman, 1996*; *Lacey and Sherman, 2017*). Although naked mole-rats exhibit high levels of within-colony relatedness (*Reeve et al., 1990*), group size also exerts a positive influence on survival, potentially generating group augmentation benefits (*O'Riain and Braude, 2001*; *le Comber et al., 2002*). Under these conditions, our model predicts the highest levels of task partitioning and division of labor. In contrast, *Neolamprologus pulcher*, a cooperatively breeding cichlid fish that lives in harsh environments with high risk of predation (*Taborsky, 2016*), shows the opposite tendency. In line with our model predictions, larger and older helpers of this species invest relatively more in territory maintenance, whereas younger/smaller helpers defend the breeding shelter of the dominant pair to a greater extent against experimentally exposed egg predators (*Bruintjes and Taborsky, 2011*). Territory maintenance has been shown to greatly affect routine metabolic rates and, hence, growth rates (*Taborsky and Grantner, 1998*), which directly translates into a decrease in the probability of becoming dominant and attaining breeding status. In addition, both group augmentation and kin selection have been found to play a role in the evolution of help in this species (*Zöttl et al., 2013b*; *García-Ruiz and Taborsky, 2022*). More research is needed to verify whether this trend is widespread in cooperatively breeding vertebrates with division of labor. The preferred form of help provided by individuals throughout their life course might deviate from our expectations if helpers exhibit differing abilities to perform distinct tasks at different ages or life stages. We assumed that individuals of all ages possessed equal capability to perform different tasks. However, as individuals age and their size and body mass increase, they may become more adept at certain tasks such as deterring predators (*Brown, 1985*; *Heinsohn and Cockburn, 1994*; *Clutton-Brock et al., 2002*; *Woxvold et al., 2006*; *Zahed et al., 2010*; *Erb and Porter, 2020*). Body condition that is unrelated to age may also influence helper contributions, as individuals in poorer condition may be more constrained on their energy allocation in activities other than self-maintenance (*Clutton-Brock et al., 2000*; *Clutton-Brock et al., 2001*; *Nichols et al., 2012*). Furthermore, our model does not capture instances in which help in general or specific tasks are enforced by the dominant pair or other more dominant group members (*Kokko et al., 2002*; *Hellmann and Hamilton, 2018*; *García-Ruiz and Taborsky, 2024*).

Despite evidence of task specialization and division of labor across several vertebrate taxa, other empirical studies have failed to find similar results. For instance, researchers failed to find evidence of task specialization in Damaraland mole-rats (*Fukomys damarensis*), despite their social structure resembling that of naked mole-rats (*Zöttl et al., 2016*; *Thorley et al., 2018*), with some variation in digging behavior with age and sex (*Rotics et al., 2025*). However, it is worth noting that researchers have only investigated correlates between different work tasks, potentially overlooking negative correlations with other helping activities that pose a more direct cost to survival. In fact, many studies that reported an absence of division of labor did not consider defensive tasks in their analyses (*English et al., 2010*; *Zöttl et al., 2016*; *Thorley et al., 2018*; *Smith et al., 2024*). Nonetheless, studies exist in which researchers did not find a negative correlation between defense and offspring provisioning or territory maintenance (*Clutton-Brock et al., 2003*; *van Asten et al., 2016*; *Teunissen et al., 2020*). However, individuals may exhibit a positive or non-relationship between tasks if they also vary in total helpfulness across their lifetime, but may still switch the proportion of time/effort allocated to different types of tasks. In our model, we addressed for this potential tendency by restricting individuals from changing their overall helpfulness with age/dominance rank. We encourage future researchers to consider the proportion of time/energy investment in addition to examining correlations between tasks.

In summary, our study helps to elucidate the potential mechanisms underlying division of labor between temporal non-reproductives via task specialization in taxa beyond eusocial organisms. Harsh environments, where individuals can obtain direct fitness benefits from group living, favor division of labor, thereby enhancing group productivity and, consequently, group size. Variation in the relative fitness costs of different helping tasks with age favors temporal polyethism, with individuals changing their behavior patterns to improve their chances of becoming breeders. Hence, future empirical research on division of labor should prioritize cooperatively breeding species inhabiting relatively harsh environments where subordinates benefit greatly from group membership, such as protection or improved food acquisition. Additionally, we emphasize the importance of including sentinel and defense behaviors in future studies to encompass a broader spectrum of fitness costs of cooperation and to investigate potential changes in task investment throughout the helpers' lifetimes.

## Materials and methods

We developed an individual-based model in which individuals may display and vary their efforts in different helping tasks. The ancestral state features an absence of alloparental care and no task specialization. We assume that residing in larger groups yields overall survival advantages, potentially resulting in direct fitness benefits associated with the increase in group size, as proposed by the group augmentation hypothesis. We also include the coevolution of the helping strategies with their dispersal propensity, as dispersal affects group size and kin structure, and thus may impact the strength of indirect fitness benefits.

## Breeding cycle

We model a population consisting of a fixed number of breeding territories, each comprising a dominant breeder that reproduces asexually and monopolizes reproduction, as well as an undefined number of subordinates that queue for the breeding position. The number of subordinates is determined by the productivity of the breeders, the dispersal decisions of the group members, and immigration. Subordinates may help increase the fecundity of the breeder, so that the breeder's fecundity depends on the cumulative level of help provided by the subordinates within the group, with an optimum increase when different helping tasks are performed to a similar degree.

Offspring inherit the genes for dispersal and helping tendencies from the breeders, potentially exhibiting slight variation in the presence of mutations (*Figure 1*, Step 1). Each genetic trait is controlled by a single locus and may take any real number. Mutations occur at each genetic locus at a frequency of $\mu$, causing minor adjustments to the allele's value inherited from the parent. This alteration involves the addition of a value drawn from a normal distribution characterized by a mean of 0 and a standard deviation of $\sigma_\mu$ (*Table 1*).

After breeders reproduce, their offspring and the subordinates in the group may disperse independently from each other. Dispersal propensity is controlled by a gene with values ranging from 0 to 1, where 0 represents certain philopatry and 1 represents absolute dispersal (*Table 1*). In each breeding cycle, dispersers may migrate to another group to become subordinates or remain as floaters waiting for breeding opportunities, which is also controlled by the same genetic dispersal predisposition as subordinates (*Figure 1*, Step 2). Successful dispersers enter the breeding queue within the group based on their dominance value rather than starting at the bottom of the hierarchy. Subordinate group members, either natal individuals born in the group or immigrants to the group, express some level of help in the form of alloparental care, which can potentially evolve to zero (*Figure 1*, Step 3). To be conservative, we use the term help as synonymous with altruistic behavior, which increases the fitness of the beneficiary (i.e. breeder) with an immediate decrease in the actor's fitness. Hence, cooperative behaviors such as defense from adult predators that incur immediate benefits to the actor as well as by-product mutualistic benefits to other group members, are not considered in this study. Dispersal propensity, alloparental care provisions, and the type of helping task they perform all depend on their genetic predisposition. They may perform tasks that reduce their body condition and, therefore, their chances of becoming breeders, such as feeding offspring or territory maintenance (hereafter called 'work tasks'), or they may perform tasks that reduce their immediate survival such as predator defense (hereafter called 'defensive tasks') (*Heinsohn and Legge, 1999*). For simplification, we assume complete dichotomous, short-term non-cumulative fitness consequences between the different types of tasks (i.e. survival and dominance costs are only sustained during the breeding season in which they perform the task, and individuals only perform one of the tasks in the given cycle). We note that the names given for each type of task are meant to facilitate interpretation, and that in some species, some tasks classified as work tasks may have a higher immediate cost in survival than in the probability to breed and vice versa. However, helping tasks should be classified in empirical systems according to the primary fitness cost associated with each type of helping activity (e.g. if a work activity mostly negatively impacts survival in a given species, this task is described in this paper as defensive).

Individual survival depends on environmental quality, group membership, group size, and the amount of help provided during defensive tasks to the breeder's offspring (*Figure 1*, Step 4). Environmental quality is expressed as a varying maximum survival probability, independent of social factors. Additional environmental effects, such as variation in mortality linked to dispersal, or on the

probability of floaters finding a new group to start breeding, have been explored elsewhere (*García-Ruiz et al., 2022*).

If the breeder in a group dies, all subordinates in the group as well as a randomly drawn sample of floaters from outside the group compete for the breeding position (*Figure 1*, Step 5). The number of floaters bidding for an empty breeding spot is then proportional to the relative abundance of floaters with regard to the total number of breeding territories, to account for spatial viscosity and the probability of being accepted in a new group (*Table 1*). In this model, age influences the probability of attaining the breeding position. Older individuals, owing to their higher RHP, have a greater probability of becoming breeders, as observed in various species (*Parker, 1974*; *Shaw, 1986*; *Doolan and Macdonald, 1996*; *Brown et al., 1997*; *Buston, 2004*; *Bridge and Field, 2007*; *Kingma et al., 2011*; *Bang and Gadagkar, 2012*; *Taborsky, 2016*; *Unnikrishnan and Gadagkar, 2021*). Age in our model corresponds to the number of breeding cycles or seasons that an individual survives. Additionally, work tasks reduce an individual's dominance value as a result of the impact on their body condition. The probability of filling an empty breeding position is then implemented as a lottery weighted by the dominance value of the candidates. Therefore, subordinates may stay in the natal group and queue to inherit the dominant position (*Woolfenden and Fitzpatrick, 1978*; *Russell and Rowley, 1993*; *Balshine-Earn et al., 1998*; *Dierkes et al., 2005*; *Bridge and Field, 2007*), but they may also disperse and breed or queue in another group (*Walters et al., 1988*; *Russell and Rowley, 1993*; *Zöttl et al., 2013a*). We ran simulations for 200,000 breeding cycles or until equilibrium was reached for all genetic traits, across 20 replicas to assess consistency.

## Age-dependent task specialization

The probability of choosing defensive versus work tasks $T$ takes a logistic function with boundaries between 0 and 1 as given in *Equation 1*, in which individuals may adjust their helping task as they age and become more dominant:

$$T = \frac{1}{1 + exp\left(\gamma_R R - \gamma_0\right)} \tag{1}$$

where $\gamma_R$ modifies the strength and direction of the effect of dominance ($R$) on task choice, and $\gamma_0$ acts as the intercept. Therefore, when $T$ approaches zero, individuals specialize in work tasks throughout their lives, and when $T$ approaches one, they specialize in defensive tasks. If $\gamma_R \neq 0$, given that $T \neq 0$ and $T \neq 1$, individuals show age-dependent task specialization.

In addition, the phenotypic expression of help ($H$) is regulated by another gene and equals the allelic value of gene $\alpha$, which remains fixed throughout the helper's life. Even though in nature individuals are likely to change the total amount of help given throughout their lives, we make this simplification to allow easier interpretation of how individuals may switch between task types as they age. How individuals in the model may adapt their level of help with age and social and environmental conditions has been described elsewhere (*García-Ruiz et al., 2022*). Hence, the gene $\alpha$ regulates the amount of help expressed, while the genes $\gamma$ determine which specific helping tasks are performed at different time points in the breeding cycle. The fitness costs of helping are described in the following section.

## Fitness costs of different helping tasks

Helping tasks are classified in our model according to their short-term fitness consequences. We assume that work tasks incur a more immediate cost to the body condition of helpers because of expenses in terms of time and energy. Work tasks reduce the dominance value $R$ to inherit the breeding position in our model following *Equation 2*, where $\gamma_h$ is a parameter that influences the strength of the cost of help ($H_{work}$) in the dominance value. We assume that dominance is influenced by age ($t$) as a proxy for RHP, with older individuals having a greater dominance value:

$$R = t - y_h H_{work} \tag{2}$$

Note that the cost to an individual's dominance value is influenced by their current age and the current specific task that they are performing, the cost of which is not cumulative over time. Therefore, an individual's effective rank within the dominance hierarchy is, irrespective of whether they belong to

a group or not, determined by their dominance value relative to other competitors for the breeding position, with higher dominance values corresponding to higher ranks in the hierarchy.

In contrast, we assume that defensive tasks have a more immediate fitness cost on survival (e.g. predator or space competitor defense) due to the high risk of injury. Therefore, subordinate helpers that defend have a survival probability ($S_H$) that decreases with the amount of help provided ($H_{defense}$) given by the logistic *Equation 3.1*, where $x_h$ is a parameter that influences the strength of the cost of help in survival:

$$S_H = \frac{1 - m}{1 + exp\left(-x_0 + x_h H_{defence} - x_n N\right)} \tag{3.1}$$

In addition to defensive tasks, social and ecological factors also influence the survival of group members. The term $m$ denotes the baseline mortality that determines how harsh the environment is. Higher values of $m$ indicate higher overall baseline mortality for all individuals in the population, irrespective of social factors. For simplicity, we assume that the harshness of the environment only impacts survival, although other effects such as on reproductive output or the strength in which help improves fecundity are likely to have an effect in nature (*Rubenstein, 2011*). Group size ($N$) provides survival benefits, such as those offered by safety in numbers or increased resource defense potential, for both subordinates and breeders (*Equations 3.1 and 3.2*, respectively), where $x_n$ is a parameter that quantifies the effect size of the benefit of group size in survival, and $x_0$ is an intercept. Floaters sustain the highest mortality ($S_F$, *Equation 3.3*) since they do not benefit from group protection:

$$S_B = \frac{1 - m}{1 + exp\left(-x_0 - x_n N\right)} \text{ and} \tag{3.2}$$

$$S_F = \frac{1 - m}{1 + exp\left(-x_0\right)} \tag{3.3}$$

## Need for division of labor

To assess the rules governing task specialization and division of labor, we first outline a basic model where individuals evolve their preferred helping task. In this model, alloparental help increases breeder productivity regardless of how tasks are distributed among helpers. We then extend this framework to a second model in which the breeder's reproductive outcome is maximized only when the group's efforts are roughly balanced across the two types of tasks, thereby actively promoting division of labor. Thus, in the second scenario, the maximum effective contribution of cumulative group help for each task type ($H_{max}$) that can enhance fecundity is given by *Equation 4*:

$$H_{max} = \frac{\sum_{i=1}^{n} H_{defense_i} + \sum_{i=1}^{n} H_{work_i}}{2} + k_m \tag{4}$$

where $k_m$ is a parameter that relaxes the requirement for perfectly balanced task allocation, allowing fecundity to increase even if the contributions to defense and work are somehow uneven. The breeders then reproduce asexually. The total number of offspring produced ($K$) is contingent upon the breeders' baseline fecundity ($k_0$) and the cumulative helping effort of all subordinates. Following *Equation 5*, fecundity increases with total help but exhibits diminishing returns:

$$K = k_0 + \frac{k_h \left(\sum_{i=1}^{n} H_{defense_i} + \sum_{i=1}^{n} H_{work_i}\right)}{1 + \sum_{i=1}^{n} H_{defense_i} + \sum_{i=1}^{n} H_{work_i}} \tag{5}$$

where $k_h$ is a parameter that scales the benefit of cumulative help on fecundity. To prevent exceeding the maximum effective help, any cumulative defense or work effort that surpasses $H_{max}$ is truncated at $H_{max}$ (i.e. if $\sum_{i=1}^{n} H_{work_i} > H_{max}$ or $\sum_{i=1}^{n} H_{defense_i} > H_{max}$). The breeder's fecundity is then determined by a randomly selected value drawn from a Poisson distribution with a mean of $K$.

## Fitness benefits of helping

We include both direct and indirect fitness benefits of helping to assess the relative importance of each in the evolution of task specialization. Indirect fitness benefits from relatedness among group members (i.e. kin selection) emerge from demographic factors such as dispersal tendencies, mortality rates, and group dynamics, though they were not coded explicitly. To calculate the coefficient of relatedness between the breeder and subordinates in a group, we calculated the coefficient of a linear regression between the allelic values of the breeders and helpers of a neutral gene that changes exclusively by genetic drift (*Michod and Hamilton, 1980*; *Gardner et al., 2011*). Direct benefits may also be obtained by enhancing group size by helping to increase the breeder's fecundity, as larger groups benefit from higher survival probabilities in our model (i.e. group augmentation).

To distinguish the effect of kin selection from group augmentation, we created two parallel models for comparison to the default model (group augmentation and kin selection acting independently), in which one of the fitness benefits was removed. In the 'only group augmentation' implementation, individuals just born (i.e. age = 1) that decide to stay in the natal territory as subordinates (*Figure 1*, Step 2) are shuffled into another group, thereby removing the influence of relatedness from the model. In the 'only kin selection' implementation, we removed the survival benefits linked to living in a group ($x_n = 0$).

## Acknowledgements

We thank Patrick Kennedy for comments on earlier versions of the manuscript and Tobias Zobrist for help with the coding implementation. Funding was provided by the SNSF grant P500PB_214371 to Irene García-Ruiz.

## Additional information

### Funding

| Funder | Grant reference number | Author |
| --- | --- | --- |
| Swiss National Science Foundation | P500PB_214371 | Irene García-Ruiz |

The funders had no role in study design, data collection and interpretation, or the decision to submit the work for publication.

### Author contributions

Irene García-Ruiz, Conceptualization, Data curation, Software, Formal analysis, Funding acquisition, Visualization, Methodology, Writing – original draft; Dustin R Rubenstein, Supervision, Validation, Writing – review and editing

### Author ORCIDs

Irene García-Ruiz ⓘ https://orcid.org/0000-0002-6139-5857
Dustin R Rubenstein ⓘ https://orcid.org/0000-0002-4999-3723

Reviewer #2 (Public review): https://doi.org/10.7554/eLife.105501.5.sa1
Author response https://doi.org/10.7554/eLife.105501.5.sa2

## Additional files

### Supplementary files
MDAR checklist

### Data availability
Custom code used to generate the simulations is available at https://github.com/IreneGR92/Task_specialization (copy archived at *García-Ruiz, 2026*).

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

# Appendix 1

## Variation in the cost of work tasks instead of defense tasks

To determine whether variation in the cost of work tasks ($y_h$) yielded qualitatively similar outcomes comparable to those observed when varying the cost of defense tasks ($x_h$), we systematically varied $y_h$ while holding $x_h$ constant (***Appendix 1—figure 1***). As predicted, our results show that an increase in cost to rank promoted individuals to choose more defense versus work tasks. Our conclusions about the role of direct and indirect fitness benefits in shaping the evolution of division of labor remain unchanged.

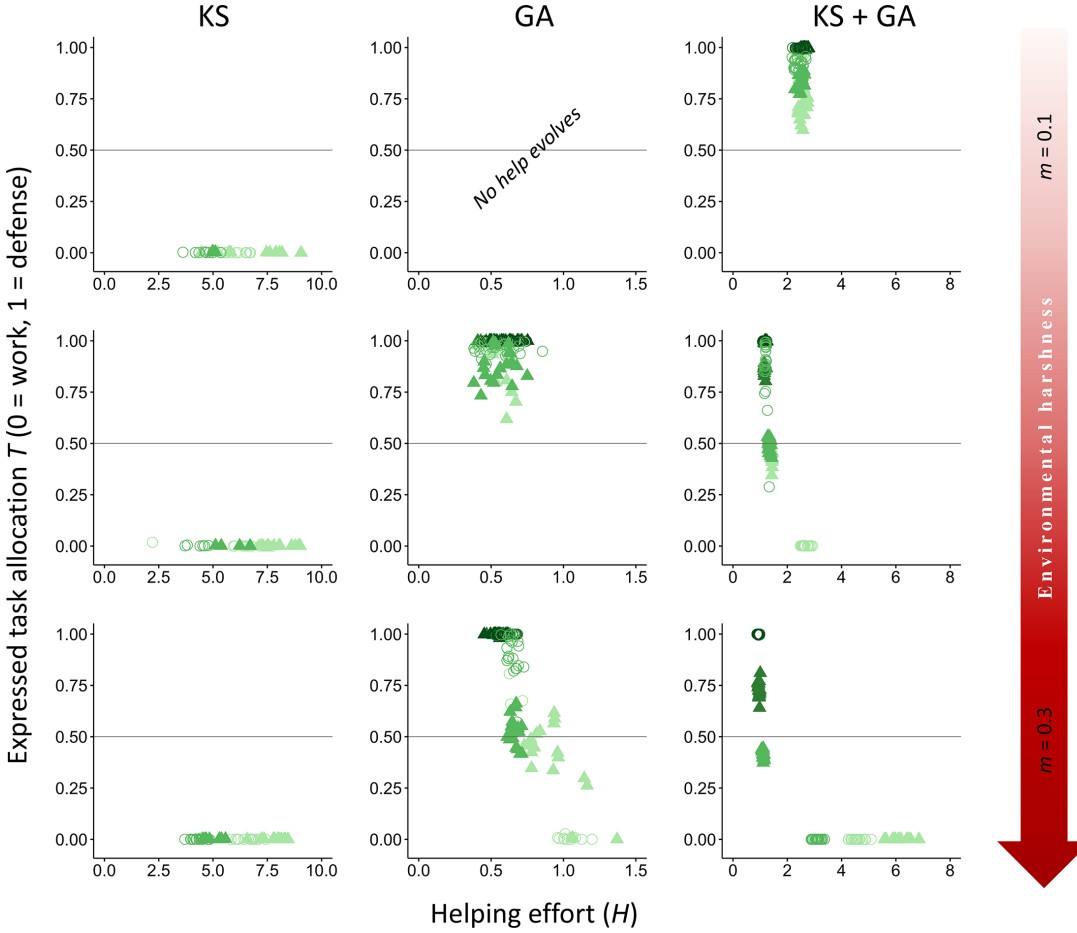

**Appendix 1—figure 1.** Effect of variation in the cost of help to rank. The evolutionary equilibria for phenotypic levels of helping and task specialization are shown at three different levels of environmental quality, ranging from benign ($m = 0.1$) to harsh ($m = 0.3$), and for three different levels of cost of help on rank (light green, $y_h = 0.05$; medium-light green, $y_h = 0.1$; medium-dark green, $y_h = 0.5$; and dark green, $y_h = 1$), across 20 replicas. The vertical axis expresses the probability of individuals choosing a defensive task with a cost to survival versus a work task with a cost to their dominance rank. The optimum breeder productivity per unit of help provided was either when both tasks were performed to a similar extent, potentially selecting for division of labor (▲), or when no restrictions were introduced to the task performed by the group members (○). In each environment, additional details are given on the selective forces that play a role in the evolution of help and task specialization: help can only evolve by kin selection (KS), group augmentation (GA), or both (KS + GA). Input parameters are the same as in ***Figure 2***, except for fixed value of $x_h = 5$ and variation in $y_h$. Additional details are provided in ***Appendix 1—table 1***.

**Appendix 1—table 1.** Supplementary data for the effect of variation in the cost of help to rank shown in **Appendix 1—figure 1**. Mean values are shown for dispersal propensity, survival probability, group size (± SD), number of floaters (± SD), ratio between helpers' and floaters' dominance value (± SD) and within-group relatedness for three environmental qualities ranging from benign (m = 0.1) to harsh (m = 0.3) across 20 replicas. Selective forces at play include kin selection (KS), group augmentation (GA), or both (KS + GA). The optimum breeder productivity per unit of help provided was either when both tasks were performed to a similar extent, potentially selecting for division of labor (DoL), or when no restrictions were introduced to the task performed by the group members (No DoL).

| | KS | | | | | | GA | | | | | | KS + GA | | | | | |
|---|---|---|---|---|---|---|---|---|---|---|---|---|---|---|---|---|---|---|
| | m = 0.1 | | m = 0.2 | | m = 0.3 | | m = 0.1 | | m = 0.2 | | m = 0.3 | | m = 0.1 | | m = 0.2 | | m = 0.3 | |
| | DoL | No DoL | DoL | No DoL | DoL | No DoL | DoL | No DoL | DoL | No DoL | DoL | No DoL | DoL | No DoL | DoL | No DoL | DoL | No DoL |
| Dispersal | 0.95 | 0.92 | 0.94 | 0.92 | 0.93 | 0.90 | 0.01 | 0.01 | 0.11 | 0.10 | 0.45 | 0.33 | 0.07 | 0.07 | 0.22 | 0.22 | 0.43 | 0.48 |
| Survival | 0.74 | 0.74 | 0.65 | 0.65 | 0.57 | 0.57 | 0.90 | 0.90 | 0.79 | 0.79 | 0.66 | 0.67 | 0.89 | 0.89 | 0.77 | 0.77 | 0.66 | 0.65 |
| Group size | 1.18±0.21 | 1.30±0.30 | 1.14±0.15 | 1.20±0.19 | 1.11±0.11 | 1.19±0.16 | 9.83±0.07 | 9.84±0.06 | 6.85±0.26 | 7.52±0.19 | 2.68±0.61 | 3.41±0.20 | 16.41±0.09 | 16.53±0.08 | 6.25±0.20 | 6.55±0.15 | 3.02±0.18 | 2.87±0.11 |
| Number of floaters | 14519±1452 | 15396±2143 | 9753±820 | 10234±1006 | 6838±421 | 7162±558 | 299±26 | 293±23 | 3514±98 | 3441±103 | 6169±548 | 5905±132 | 5930±128 | 5982±107 | 7438±158 | 7774±258 | 7594±1560 | 8704±1139 |
| Rank ratio helpers vs floaters | 0.80±0.20 | 0.83±0.20 | 0.77±0.25 | 0.71±0.35 | 0.77±0.31 | 0.66±0.33 | 1.05±0.06 | 1.05±0.05 | 1.05±0.02 | 1.05±0.03 | 1.03±0.03 | 1.02±0.04 | 1.02±0.02 | 1.03±0.02 | 1.00±0.02 | 1.00±0.02 | 0.94±0.05 | 0.95±0.06 |
| Relatedness | 0.38 | 0.36 | 0.53 | 0.52 | 0.74 | 0.72 | 0.00 | 0.00 | 0.00 | 0.00 | 0.00 | 0.00 | 0.56 | 0.57 | 0.57 | 0.57 | 0.64 | 0.62 |

## Appendix 2

### Kin selection and the evolution of division of labor

To assess the robustness of our findings on the evolution of division of labor driven by kin selection independent of group survival benefits, we further examined the influence of survival, dispersal, relatedness, and breeding opportunities outside the natal group.

One factor contributing to defense not readily evolving under kin selection alone is that the cost to dominance is relatively small when competing against relatives. Another possible explanation is that without group benefits, mortality rates are too high for survival-costly helping tasks such as defense to evolve. In *Appendix 2—figure 1*, we examine the effects of increasing the baseline survival from $x_0 = 1.5$ (as in *Figure 2*) to $x_0 = 4.5$ (which simulates the addition of a group benefit of $x_n = 3$ to the baseline survival) and $x_0 = 10$ (an extreme scenario that greatly reduces the survival cost of defense tasks; see *Equation 3.1*).

Our results indicate that even with high baseline survival, kin selection strongly favors work over defense tasks. However, in very harsh environments ($m=0.3$), where the survival costs of defense tasks are minimal ($x_0 = 10$ and $x_h = 3$; *Appendix 2—table 1*), some degree of task specialization and division of labor emerges (*Appendix 2—figure 1*). Despite this, most individuals still choose to disperse and breed outside their natal territory when no group survival benefits exist (*Appendix 2—table 1*). This suggests that, without such benefits, offspring have little incentive to stay and help due to reduced reproductive opportunities and competition with kin. To test this, we eliminated the dispersal advantage by reducing the number of sampled groups by dispersers from two ($f = 2$) to one ($f = 1$), incentivizing individuals to queue for breeding in their natal territory (*Appendix 2—figure 2*).

Even when increased philopatry, resulting from reduced outbreeding opportunities for floaters, potentially enhances the indirect fitness benefits of helping and task partitioning, division of labor only evolves when survival costs are exceedingly low ($x_0 = 10$; *Appendix 2—figure 2*; *Appendix 2—table 2*). This suggests that while kin selection can promote the evolution of division of labor, it does so only under highly restrictive conditions with minimal survival costs. Therefore, in natural settings, other selective forces, such as direct fitness benefits, are more likely to drive the evolution of division of labor.

The strength of kin selection is likely more pronounced in our model than in natural populations due to the assumption of asexual reproduction. To assess whether the results remain valid with lower within-group relatedness, we shuffled half of the newly born individuals (i.e. age = 1) who choose to stay in their natal territory as subordinates, similar to the '*only group augmentation*' implementation. The results indicate that a reduction in relatedness increases the probability of individuals favoring defense over work tasks, but the overall conclusions of our model remain unchanged (*Appendix 2—figure 3*).

So far, we have considered scenarios where alloparental care always increases breeder productivity, regardless of task partitioning. Because we assume parents provide parental care, reproduction is possible even without help, but productivity is maximized when helpers contributed similarly to both tasks, promoting rather than requiring division of labor. To explore whether kin selection alone can favor division of labor in situations where help boosts productivity *only* if all tasks are performed to a similar extent, we developed a new implementation of the model. In this implementation, the absence of task division prevents any productivity gains, thereby making division of labor obligatory. To achieve this, instead of relying on $H_{max}$ (*Equation 4*), we introduce an $H_{min}$

$$H_{min} = \left| \sum_{i=1}^{n} H_{defense_i} - \sum_{i=1}^{n} H_{work_i} \right| \tag{A1}$$

and compare it to the maximum values of the cumulative help for each task, allowing a proportional difference $k_p$ from the mean. Hence, if

$$H_{min} > k_p \cdot \max \left( \sum_{i=1}^{n} H_{defense_i} , \sum_{i=1}^{n} H_{work_i} \right) \tag{A2}$$

then $K = k_0$, where costs associated with help performance are still inflicted to the actor. The results of this implantation of the model show that help and division of labor, as well as philopatry, do evolve only under kin selection benefits only if the increase in productivity per unit of help is very high (*Appendix 2—figure 4*). Therefore, kin selection alone can promote division of labor only under restrictive conditions: when alloparental care produces a substantial increase in productivity and when both types of tasks are required in similar measure for successful offspring production and survival. These constraints suggest that grouping benefits are more likely to play a key role in the emergence of division of labor. In addition, due to the need for coordination between group members, division of labor only emerged when we assumed a philopatric starting point ($\beta_{init} = 0.5$).

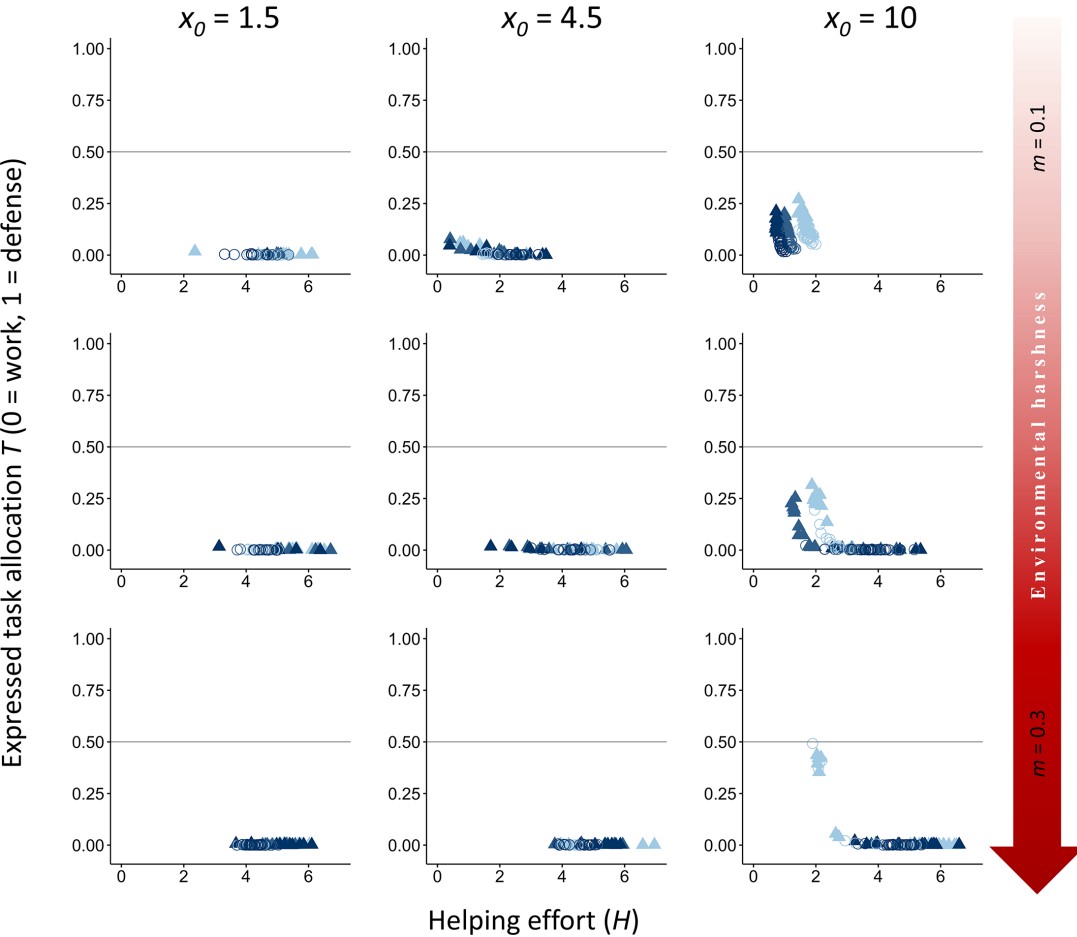

**Appendix 2—figure 1.** Effect of increasing the baseline survival $x_0$ to favor the evolution of division of labor under only kin selection. The evolutionary equilibria for levels of helping and task specialization are shown at three different levels of environmental quality, ranging from benign ($m = 0.1$) to harsh ($m = 0.3$), and for three different levels of cost of help on survival (light blue, $x_h = 3$; blue, $x_h = 5$; and dark blue, $x_h = 7$). The vertical axis represents the expressed task allocation between a defensive task with a cost to survival versus a work task with a cost to dominance. The optimum breeder productivity per unit of help provided was either when both tasks were performed to a similar extent, potentially selecting for division of labor (▲), or when no restrictions were introduced to the task performed by the group members (○). Input parameters are the same as in *Figure 2* (where $x_0 = 1.5$) except for $x_0 = 4.5$ and $x_0 = 10$ (higher survival for all individuals irrespective of group membership or environment; *Table 1*). Additional details are provided in *Appendix 2—table 1*.

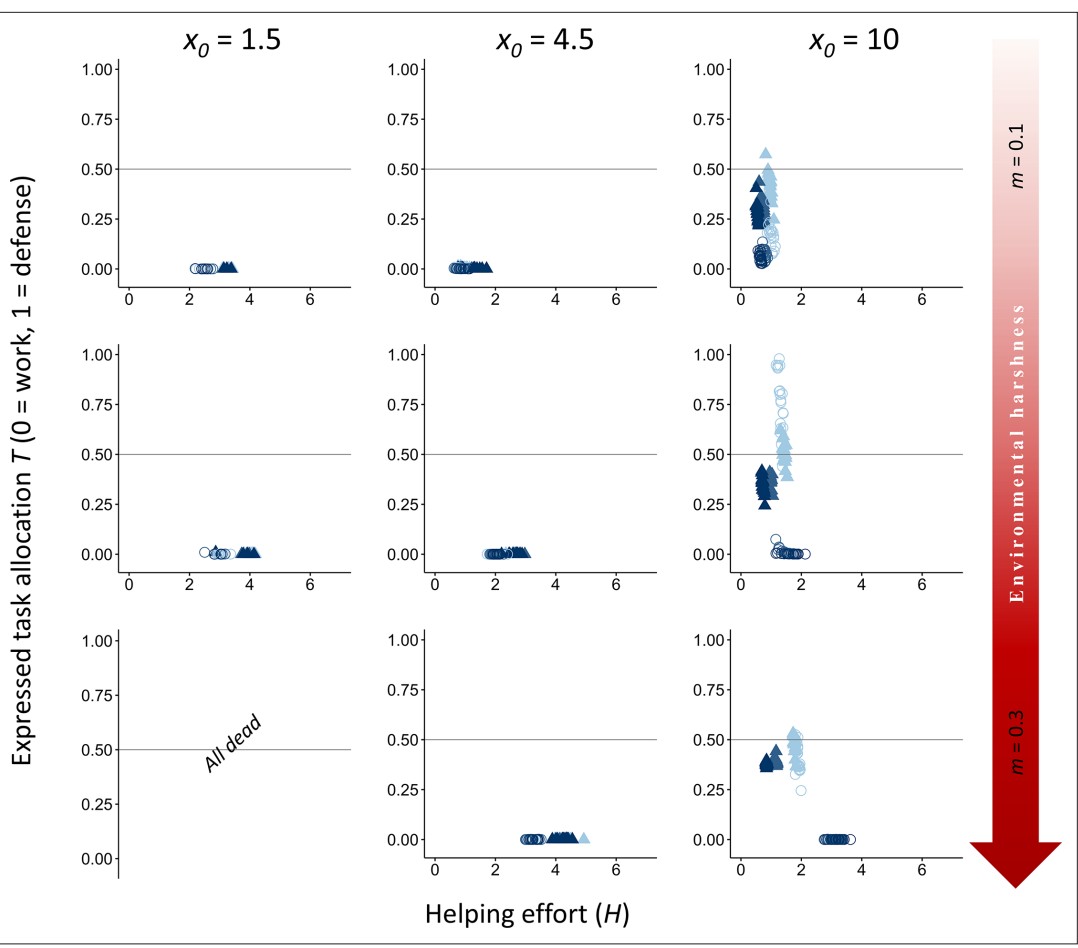

**Appendix 2—figure 2.** Effect of reducing the incentives to disperse to favor the evolution of division of labor under only kin selection. An increase in incentives to remain philopatric was achieved by reducing $f$ to 1 ($f = 2$ in *Figure 2*; *Table 1*). The evolutionary equilibria for levels of helping and task specialization are shown at three different levels of environment quality that range from benign ($m = 0.1$) to harsh ($m = 0.3$), and for three different levels of cost of help on survival (light blue, $x_h = 3$; blue, $x_h = 5$; and dark blue, $x_h = 7$). The vertical axis represents the expressed task allocation between a defensive task with a cost to survival versus a work task with a cost to dominance. The optimum breeder productivity per unit of help provided was either when both tasks were performed to a similar extent, potentially selecting for division of labor (▲), or when no restrictions were introduced to the task performed by the group members (○). Other input parameters are the same as in *Figure 2* (where $x_0 = 1.5$) except for $x_0 = 4.5$ and $x_0 = 10$ (higher survival for all individuals irrespective of group membership or environment; *Table 1*). Additional details are provided in *Appendix 2—table 2*.

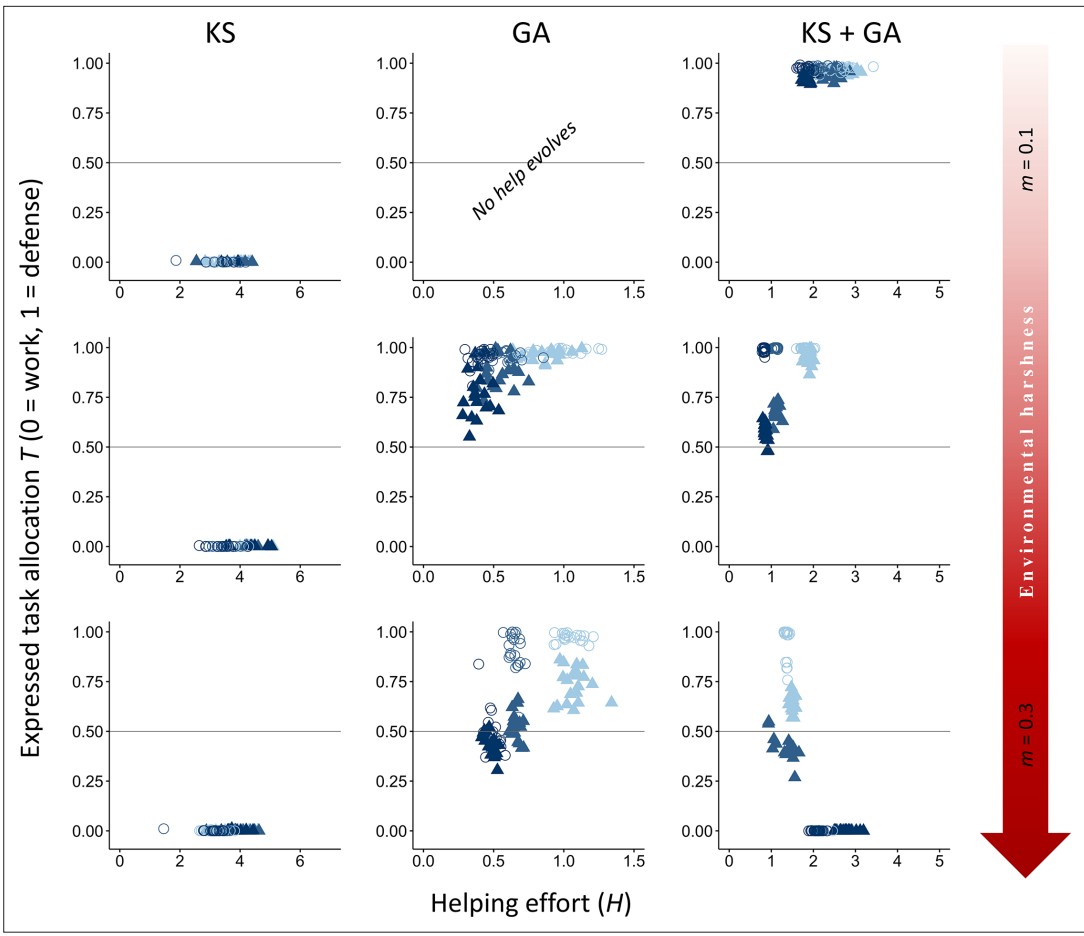

**Appendix 2—figure 3.** Effect of reducing within-group relatedness by half to mimic sexual reproduction. A reduction in within-group relatedness was achieved by shuffling half of the philopatric newborns to another group in the kin selection (KS) and KS+ group augmentation (GA) implementations, and all for the GA implementation. The evolutionary equilibria for levels of helping and task specialization are shown at three different levels of environment quality that range from benign ($m = 0.1$) to harsh ($m = 0.3$), and for three different levels of cost of help on survival (light blue, $x_h = 3$; blue, $x_h = 5$; and dark blue, $x_h = 7$). The vertical axis represents the expressed task allocation between a defensive task with a cost to survival versus a work task with a cost to dominance. The optimum breeder productivity per unit of help provided was either when both tasks were performed to a similar extent, potentially selecting for division of labor (▲), or when no restrictions were introduced to the task performed by the group members (○). Input parameters are the same as in *Figure 2*. Additional details are provided in *Appendix 2—table 3*.

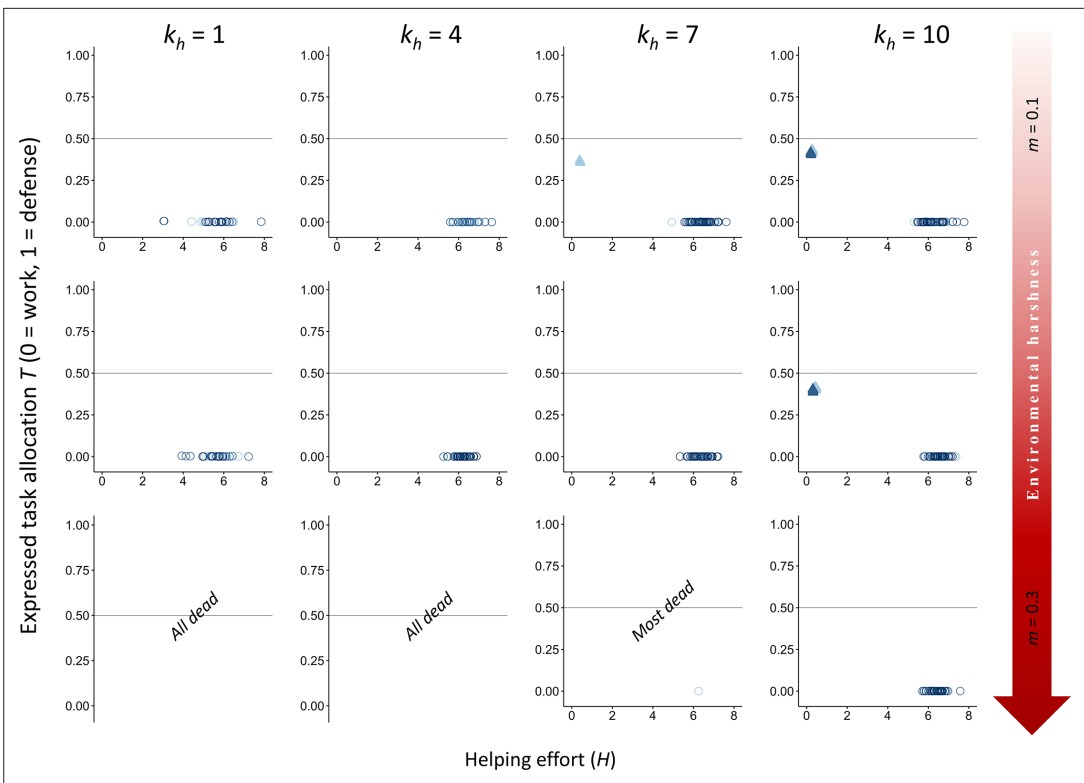

**Appendix 2—figure 4.** Effect of obligatory division of labor in the emergence of task division under only kin selection fitness benefits. For an increase in the productivity of the breeder, both kinds of task had to be performed to a similar extent at the group level. The evolutionary equilibria for levels of helping and task specialization are shown at three different levels of environment quality that range from benign ($m = 0.1$) to harsh ($m = 0.3$), and for three different levels of cost of help on survival (light blue, $x_h = 3$; blue, $x_h = 5$; and dark blue, $x_h = 7$). The vertical axis represents the expressed task allocation between a defensive tasks with a cost to survival versus a work tasks with a cost to dominance. Breeder productivity was enhanced *exclusively* when both tasks were performed to a similar extent (obligatory division of labor; ▲), or simply when the task performed similarly by the group members (favorable division of labor; ○). Input parameters are the same as in *Figure 2* except for variation in $k_h$ ($k_h = 1$ is default), $\beta_{init} = 0.5$, and $k_p = 0.5$. Additional details are provided in *Appendix 2—table 4*.

**Appendix 2—table 1.** Supplementary data for the effect of increasing the baseline survival $x_0$ to favor the evolution of division of labor under only kin selection shown in *Appendix 2—figure 1*.

Mean values are shown for dispersal propensity, survival probability, group size (± SD), number of floaters (± SD), ratio between helpers' and floaters' dominance value (± SD) and within-group relatedness for three environmental qualities ranging from benign ($m = 0.1$) to harsh ($m = 0.3$) across 20 replicas. Results are shown for $x_0 = 1.5$ (default), $x_0 = 3.5$, and $x_0 = 10$. The optimum breeder productivity per unit of help provided was either when both tasks were performed to a similar extent, potentially selecting for division of labor (DoL), or when no restrictions were introduced to the task performed by the group members (No DoL).

| | $x_0 = 1.5$ | | | | | | $x_0 = 4.5$ | | | | | | $x_0 = 10$ | | | | | |
| | $m = 0.1$ | | $m = 0.2$ | | $m = 0.3$ | | $m = 0.1$ | | $m = 0.2$ | | $m = 0.3$ | | $m = 0.1$ | | $m = 0.2$ | | $m = 0.3$ | |
| | DoL | No DoL | DoL | No DoL | DoL | No DoL | DoL | No DoL | DoL | No DoL | DoL | No DoL | DoL | No DoL | DoL | No DoL | DoL | No DoL |
|---|---|---|---|---|---|---|---|---|---|---|---|---|---|---|---|---|---|---|
| Dispersal | 0.95 | 0.92 | 0.94 | 0.90 | 0.91 | 0.84 | 0.84 | 0.85 | 0.89 | 0.86 | 0.89 | 0.85 | 0.79 | 0.83 | 0.88 | 0.85 | 0.9 | 0.85 |
| Survival | 0.74 | 0.74 | 0.65 | 0.65 | 0.57 | 0.57 | 0.89 | 0.89 | 0.79 | 0.79 | 0.69 | 0.69 | 0.90 | 0.90 | 0.80 | 0.80 | 0.70 | 0.70 |
| Group size | 1.16±0.16 | 1.34±0.30 | 1.14±0.12 | 1.25±0.19 | 1.14±0.08 | 1.29±0.11 | 3.02±0.33 | 3.14±0.14 | 1.61±0.26 | 1.86±0.21 | 1.33±0.13 | 1.50±0.09 | 4.19±0.19 | 3.75±0.20 | 1.68±0.22 | 1.96±0.07 | 1.31±0.11 | 1.49±0.07 |
| Number of floaters | 14280±1352 | 15616±2322 | 9778±711 | 10352±1082 | 6956±330 | 7482±395 | 53241±5074 | 61820±1393 | 22911±2167 | 25655±1903 | 12672±836 | 13848±548 | 59534±3345 | 65170±2536 | 23585±1799 | 27291±560 | 12901±775 | 14293±574 |
| Rank ratio helpers vs floaters | 0.91±0.27 | 0.94±0.22 | 0.85±0.24 | 0.88±0.25 | 0.76±0.27 | 0.75±0.12 | 0.97±0.06 | 0.95±0.09 | 0.90±0.16 | 0.85±0.11 | 0.81±0.12 | 0.85±0.09 | 1.00±0.05 | 0.99±0.06 | 0.98±0.13 | 0.93±0.07 | 0.89±0.19 | 0.84±0.13 |
| Relatedness | 0.37 | 0.36 | 0.54 | 0.52 | 0.73 | 0.71 | 0.15 | 0.15 | 0.28 | 0.28 | 0.45 | 0.44 | 0.17 | 0.15 | 0.27 | 0.27 | 0.44 | 0.43 |

**Appendix 2—table 2.** Supplementary data for the effect of reducing the incentives to disperse to favor the evolution of division of labor under only kin selection shown in *Appendix 2—figure 2*.

The increased incentive was achieved by reducing the parameter $f$ that signifies the mean number of groups a floater samples for becoming a breeder from 2 (default) to 1. Mean values are shown for dispersal propensity, survival probability, group size ($\pm$ SD), number of floaters ($\pm$ SD), ratio between helpers' and floaters' dominance value ($\pm$ SD) and within-group relatedness for three environmental qualities ranging from benign ($m = 0.1$) to harsh ($m = 0.3$) across 20 replicas. Results are shown for $x_0 = 1.5$ (default), $x_0 = 3.5$, and $x_0 = 10$. The optimum breeder productivity per unit of help provided was either when both tasks were performed to a similar extent, potentially selecting for division of labor (DoL), or when no restrictions were introduced to the task performed by the group members (No DoL).

| | $x_0 = 1.5$ | | | | | | $x_0 = 4.5$ | | | | | | $x_0 = 10$ | | | | | |
| | $m = 0.1$ | | $m = 0.2$ | | $m = 0.3$ | | $m = 0.1$ | | $m = 0.2$ | | $m = 0.3$ | | $m = 0.1$ | | $m = 0.2$ | | $m = 0.3$ | |
| | DoL | No DoL | DoL | No DoL | DoL | No DoL | DoL | No DoL | DoL | No DoL | DoL | No DoL | DoL | No DoL | DoL | No DoL | DoL | No DoL |
|---|---|---|---|---|---|---|---|---|---|---|---|---|---|---|---|---|---|---|
| Dispersal | 0.82 | 0.75 | 0.79 | 0.79 | NA | NA | 0.23 | 0.24 | 0.49 | 0.50 | 0.72 | 0.68 | 0.25 | 0.25 | 0.44 | 0.47 | 0.49 | 0.66 |
| Survival | 0.74 | 0.74 | 0.65 | 0.65 | 0 | 0 | 0.89 | 0.89 | 0.79 | 0.79 | 0.69 | 0.69 | 0.90 | 0.90 | 0.80 | 0.80 | 0.70 | 0.70 |
| Group size | 1.90±1.07 | 2.34±1.26 | 1.71±0.62 | 1.75±0.75 | 0 | 0 | 13.09±0.23 | 13.35±0.40 | 4.84±0.38 | 4.90±0.08 | 2.04±0.06 | 2.33±0.05 | 14.29±0.26 | 14.59±0.16 | 5.17±0.37 | 5.30±0.08 | 3.10±0.26 | 2.47±0.05 |
| Number of floaters | 14416±972 | 14468±1058 | 9986±773 | 9957±859 | 0 | 0 | 18182±894 | 19426±1767 | 18440±1678 | 19531±337 | 13682±154 | 14236±185 | 21746±639 | 22100±361 | 16541±1838 | 18755±393 | 9839±265 | 14274±515 |
| Rank ratio helpers vs floaters | 1.10±0.43 | 1.01±0.19 | 0.96±0.37 | 1.04±0.59 | NA | NA | 0.98±0.03 | 0.98±0.03 | 0.93±0.04 | 0.96±0.04 | 0.86±0.09 | 0.86±0.07 | 0.99±0.02 | 0.98±0.03 | 0.98±0.05 | 0.97±0.04 | 1.00±0.05 | 0.89±0.08 |
| Relatedness | 0.40 | 0.39 | 0.55 | 0.56 | NA | NA | 0.32 | 0.32 | 0.37 | 0.36 | 0.46 | 0.46 | 0.29 | 0.29 | 0.38 | 0.36 | 0.52 | 0.46 |

**Appendix 2—table 3.** Supplementary data for effect of reducing within-group relatedness by half to mimic sexual reproduction shown in *Appendix 2—figure 3*. Mean values are shown for dispersal propensity, survival probability, group size (± SD), number of floaters (± SD), ratio between helpers' and floaters' dominance value (± SD) and within-group relatedness for three environmental qualities ranging from benign ($m = 0.1$) to harsh ($m = 0.3$) across 20 replicas. Selective forces at play include kin selection (KS), group augmentation (GA), or both (KS + GA). The optimum breeder productivity per unit of help provided was either when both tasks were performed to a similar extent, potentially selecting for division of labor (DoL), or when no restrictions were introduced to the task performed by the group members (No DoL).

| | KS | | | | | | GA | | | | | | KS + GA | | | | | |
| | $m = 0.1$ | | $m = 0.2$ | | $m = 0.3$ | | $m = 0.1$ | | $m = 0.2$ | | $m = 0.3$ | | $m = 0.1$ | | $m = 0.2$ | | $m = 0.3$ | |
|---|---|---|---|---|---|---|---|---|---|---|---|---|---|---|---|---|---|---|
| | DoL | No DoL | DoL | No DoL | DoL | No DoL | DoL | No DoL | DoL | No DoL | DoL | No DoL | DoL | No DoL | DoL | No DoL | DoL | No DoL |
| Dispersal | 0.94 | 0.93 | 0.93 | 0.90 | 0.91 | 0.88 | 0.01 | 0.01 | 0.11 | 0.10 | 0.37 | 0.33 | 0.02 | 0.02 | 0.11 | 0.11 | 0.41 | 0.50 |
| Survival | 0.74 | 0.74 | 0.65 | 0.65 | 0.57 | 0.57 | 0.90 | 0.90 | 0.79 | 0.79 | 0.66 | 0.67 | 0.90 | 0.90 | 0.79 | 0.79 | 0.66 | 0.65 |
| Group size | 1.20±0.19 | 1.29±0.27 | 1.16±0.12 | 1.26±0.18 | 1.14±0.08 | 1.22±0.14 | 9.85±0.08 | 9.83±0.05 | 6.94±0.43 | 7.52±0.41 | 3.04±0.35 | 3.40±0.36 | 18.37±0.10 | 18.54±0.09 | 7.66±0.15 | 7.86±0.21 | 3.16±0.53 | 2.78±0.09 |
| Number of floaters | 14491±1329 | 15139±2030 | 9820±682 | 10300±1004 | 8869±291 | 7123±486 | 297±23 | 297±22 | 3512±104 | 3428±101 | 5925±137 | 5905±122 | 2216±74 | 2218±83 | 4164±124 | 4179±119 | 7327±1378 | 9020±346 |
| Rank ratio helpers vs floaters | 0.90±0.24 | 0.98±0.26 | 0.90±0.27 | 0.91±0.22 | 0.85±0.26 | 0.93±0.25 | 1.05±0.06 | 1.06±0.06 | 1.05±0.04 | 1.06±0.04 | 1.03±0.06 | 1.03±0.06 | 1.06±0.03 | 1.06±0.03 | 1.02±0.04 | 1.03±0.04 | 0.95±0.07 | 0.94±0.06 |
| Relatedness | 0.18 | 0.20 | 0.27 | 0.26 | 0.36 | 0.36 | 0 | 0 | 0 | 0 | 0 | 0 | 0.30 | 0.31 | 0.31 | 0.31 | 0.32 | 0.30 |

**Appendix 2—table 4.** Supplementary data for the effect of obligatory division of labor in the emergence of task division under only kin selection fitness benefits shown in **Appendix 2—figure 4.**

Mean values are shown for dispersal propensity, survival probability, group size (± SD), number of floaters (± SD), ratio between helpers' and floaters' dominance value (± SD), and within-group relatedness for three environmental qualities ranging from benign ($m = 0.1$) to harsh ($m = 0.3$) across 20 replicas. The optimum breeder productivity per unit of help provided was either when both tasks were performed to a similar extent, potentially selecting for division of labor (DoL favored), or when division of labor was required to increase the breeder's productivity (DoL obligatory). Note that for $k_h = 7$, $m = 0.3$ implementation 'DoL favored', only one replicate survived, hence SD is not defined.

**$k_h = 1$**

| | m = 0.1 | | m = 0.2 | | m = 0.3 | |
| --- | --- | --- | --- | --- | --- | --- |
| | DoL obligatory | DoL favored | DoL obligatory | DoL favored | DoL obligatory | DoL favored |
| Dispersal | 0.97 | 0.92 | 0.97 | 0.91 | NA | NA |
| Survival | 0.74 | 0.74 | 0.65 | 0.65 | 0 | 0 |
| Group size | 1.07±0.01 | 1.29±0.20 | 1.05±0.01 | 1.21±0.12 | NA | NA |
| Number of floaters | 13560±127 | 15323±1661 | 9207±133 | 10168±728 | NA | NA |
| Rank ratio helpers vs floaters | 1.01±0.05 | 0.91±0.11 | 1.01±0.05 | 0.84±0.15 | NA | NA |
| Relatedness | 0.37 | 0.37 | 0.53 | 0.52 | NA | NA |

**$k_h = 7$**

| | m = 0.1 | | m = 0.2 | | m = 0.3 | |
| --- | --- | --- | --- | --- | --- | --- |
| | DoL obligatory | DoL favored | DoL obligatory | DoL favored | DoL obligatory | DoL favored |
| Dispersal | 0.95 | 0.87 | 0.97 | 0.85 | NA | 0.83 |
| Survival | 0.73 | 0.74 | 0.65 | 0.65 | 0 | 0.57 |
| Group size | 1.24±0.65 | 4.36±0.14 | 1.05±0.01 | 3.81±0.11 | NA | 3.29 |
| Number of floaters | 13835±995 | 110167±935 | 9204±133 | 77579±642 | NA | 56608 |
| Rank ratio helpers vs floaters | 1.00±0.04 | 0.82±0.02 | 1.01±0.04 | 0.77±0.02 | NA | 0.69 |
| Relatedness | 0.39 | 0.31 | 0.54 | 0.4 | NA | 0.48 |

**$k_h = 4$**

| | m = 0.1 | | m = 0.2 | | m = 0.3 | |
| --- | --- | --- | --- | --- | --- | --- |
| | DoL obligatory | DoL favored | DoL obligatory | DoL favored | DoL obligatory | DoL favored |
| Dispersal | 0.97 | 0.83 | 0.97 | 0.8 | NA | NA |
| Survival | 0.74 | 0.74 | 0.65 | 0.65 | 0 | 0 |
| Group size | 1.07±0.01 | 3.42±0.09 | 1.05±0.01 | 3.02±0.07 | NA | NA |
| Number of floaters | 13591±134 | 60761±501 | 9201±127 | 40928±451 | NA | NA |
| Rank ratio helpers vs floaters | 1.00±0.04 | 0.82±0.02 | 1.01±0.05 | 0.76±0.01 | NA | NA |
| Relatedness | 0.37 | 0.32 | 0.54 | 0.42 | NA | NA |

**$k_h = 10$**

| | m = 0.1 | | m = 0.2 | | m = 0.3 | |
| --- | --- | --- | --- | --- | --- | --- |
| | DoL obligatory | DoL favored | DoL obligatory | DoL favored | DoL obligatory | DoL favored |
| Dispersal | 0.61 | 0.89 | 0.67 | 0.87 | NA | 0.86 |
| Survival | 0.71 | 0.74 | 0.63 | 0.65 | NA | 0.57 |
| Group size | 5.62±3.31 | 5.01±0.21 | 3.00±1.43 | 4.36±0.15 | NA | 3.9±0.10 |
| Number of floaters | 21631±6556 | 161169±1227 | 13833±4002 | 115694±859 | NA | 87503±642 |
| Rank ratio helpers vs floaters | 1.00±0.03 | 0.83±0.02 | 1.00±0.03 | 0.76±0.01 | NA | 0.71±0.02 |
| Relatedness | 0.42 | 0.29 | 0.55 | 0.39 | NA | 0.47 |

## Appendix 3

### Dominance-dependent dispersal propensities

To assess the role of dominance or RHP on dispersal propensities $D$, and its potential effect on task specialization, we added a reaction norm to dominance value ($R$):

$$D = \frac{1}{1 + exp\left(-\beta_R R - \beta_0\right)} \tag{A3}$$

where $\beta_R$ modifies the strength and direction of the effect of dominance on dispersal, and $\beta_0$ acts as the intercept. This change allows individuals to adjust their dispersal and immigration propensities to their competitiveness, as well as their preferred helping tasks.

We found that incorporating a dispersal reaction norm to rank did not qualitatively influence the results on the evolution of division of labor and task specialization (**Appendix 3—figure 1**), with the conclusions remaining broadly unaltered.

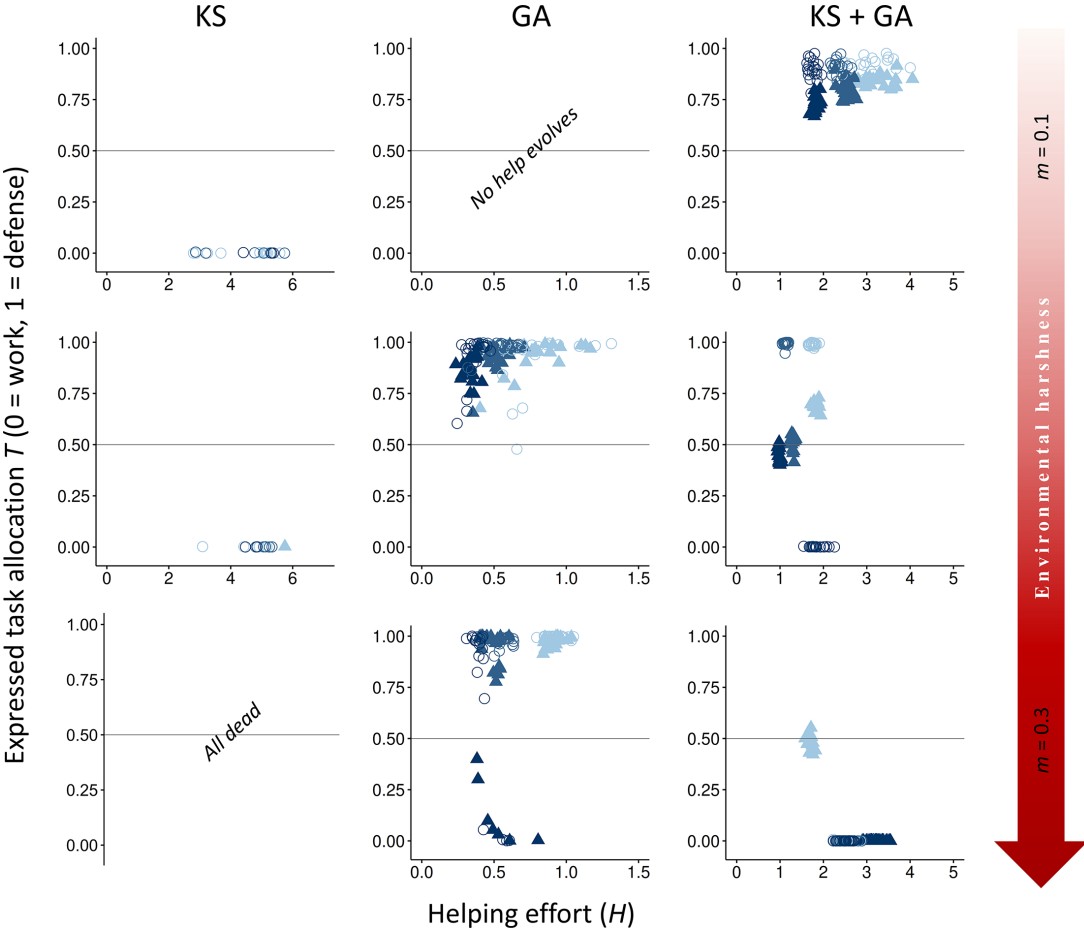

**Appendix 3—figure 1.** Effect of adding a reaction norm of dispersal and immigration propensity to dominance value. The evolutionary equilibria for phenotypic levels of helping and task specialization are shown at three different levels of environmental quality, ranging from benign ($m = 0.1$) to harsh ($m = 0.3$), and for three different levels of cost of help on survival (light blue, $x_h = 3$; blue, $x_h = 5$; and dark blue, $x_h = 7$), across 20 replicas. The vertical axis represents the expressed task allocation between a defensive task with a cost to survival versus a work task with a cost to their dominance rank. The optimum breeder productivity per unit of help provided was either when both tasks were performed to a similar extent, potentially selecting for division of labor (▲), or when no restrictions were introduced to the task performed by the group members (○). In each environment, additional details are given on the selective forces that play a role in the evolution of help and task specialization: help can only evolve by kin selection (KS), group augmentation (GA), or both (KS + GA). Input parameters are the same as in **Figure 2** with the addition of the $\beta_0$ and $\beta_R$. Additional details are provided in **Appendix 3—table 1**.

**Appendix 3—table 1.** Supplementary data for the effect of adding a reaction norm of dispersal and immigration propensity to dominance value shown in *Appendix 3—figure 1*.

Mean values are shown for dispersal propensity, survival probability, group size (± SD), number of floaters (± SD), ratio between helpers' and floaters' dominance value (± SD) and within-group relatedness for three environmental qualities ranging from benign ($m = 0.1$) to harsh ($m = 0.3$) across 20 replicas. Selective forces at play include kin selection (KS), group augmentation (GA), or both (KS + GA). The optimum breeder productivity per unit of help provided was either when both tasks were performed to a similar extent, potentially selecting for division of labor (DoL), or when no restrictions were introduced to the task performed by the group members (No DoL).

| | KS | | | | | | GA | | | | | | KS+GA | | | | | |
|---|---|---|---|---|---|---|---|---|---|---|---|---|---|---|---|---|---|---|
| | $m = 0.1$ | | $m = 0.2$ | | $m = 0.3$ | | $m = 0.1$ | | $m = 0.2$ | | $m = 0.3$ | | $m = 0.1$ | | $m = 0.2$ | | $m = 0.3$ | |
| | DoL | No DoL | DoL | No DoL | DoL | No DoL | DoL | No DoL | DoL | No DoL | DoL | No DoL | DoL | No DoL | DoL | No DoL | DoL | No DoL |
| Dispersal | 1.00 | 0.94 | 1.00 | 0.96 | NA | NA | 0.00 | 0.00 | 0.16 | 0.15 | 0.67 | 0.51 | 0.08 | 0.08 | 0.20 | 0.20 | 0.43 | 0.45 |
| $\beta_R$ | −3.71 | −3.17 | −4.27 | −4.18 | NA | NA | 3.99 | 3.99 | −0.30 | −0.48 | −4.61 | −4.52 | 2.71 | 2.72 | 0.67 | 2.17 | 2.45 | 3.49 |
| Survival | 0.74 | 0.74 | 0.65 | 0.65 | NA | NA | 0.90 | 0.90 | 0.78 | 0.78 | 0.65 | 0.64 | 0.89 | 0.89 | 0.77 | 0.77 | 0.66 | 0.65 |
| Group size | 1.00±0.00 | 1.27±0.39 | 1.01±0.06 | 1.12±0.22 | NA | NA | 9.76±0.08 | 9.75±0.11 | 6.15±0.78 | 6.77±0.99 | 1.87±0.57 | 2.44±0.16 | 16.21±0.10 | 16.23±0.10 | 6.60±0.16 | 6.85±0.17 | 3.13±0.38 | 3.15±0.06 |
| Number of floaters | 13928±123 | 15486±2148 | 9477±236 | 9943±928 | NA | NA | 25±9 | 26±9 | 4630±493 | 4740±1303 | 7350±504 | 7367±730 | 6408±148 | 6537±144 | 7059±188 | 7190±177 | 8055±1161 | 8935±150 |
| Rank ratio helpers vs floaters | 0.47±0.16 | 0.98±1.18 | 0.58±0.27 | 0.45±0.05 | NA | NA | 5.70±0.43 | 5.76±0.44 | 0.55±0.06 | 0.51±0.07 | 0.62±0.12 | 0.55±0.02 | 5.91±0.19 | 6.07±0.18 | 1.83±0.45 | 2.43±0.48 | 1.91±0.17 | 2.06±0.12 |
| Relatedness | 0.31 | 0.34 | 0.31 | 0.30 | NA | NA | 0 | 0 | 0 | 0 | 0 | 0 | 0.37 | 0.37 | 0.48 | 0.43 | 0.42 | 0.36 |

## Appendix 4

## Variation in the benefit of group augmentation

To determine whether division of labor is maintained when reducing group augmentation benefits, we reduced $x_n$ from 3 (*Figure 2*) to 1 (*Appendix 4—figure 1*). When group augmentation benefits to survival are low, defense behaviors mainly evolve in benign environments with low mortality risk when both indirect and direct benefits take effect. However, defense can also evolve under harsher conditions when kinship effects are negligible. In contrast, reducing the cost of defense ($x_n = 1$, *Appendix 4—figure 1*) did not facilitate the evolution of defense behavior when only kin selection is acting. This pattern persists even at very low defense costs (e.g. $x_n = 0.1$, not shown).

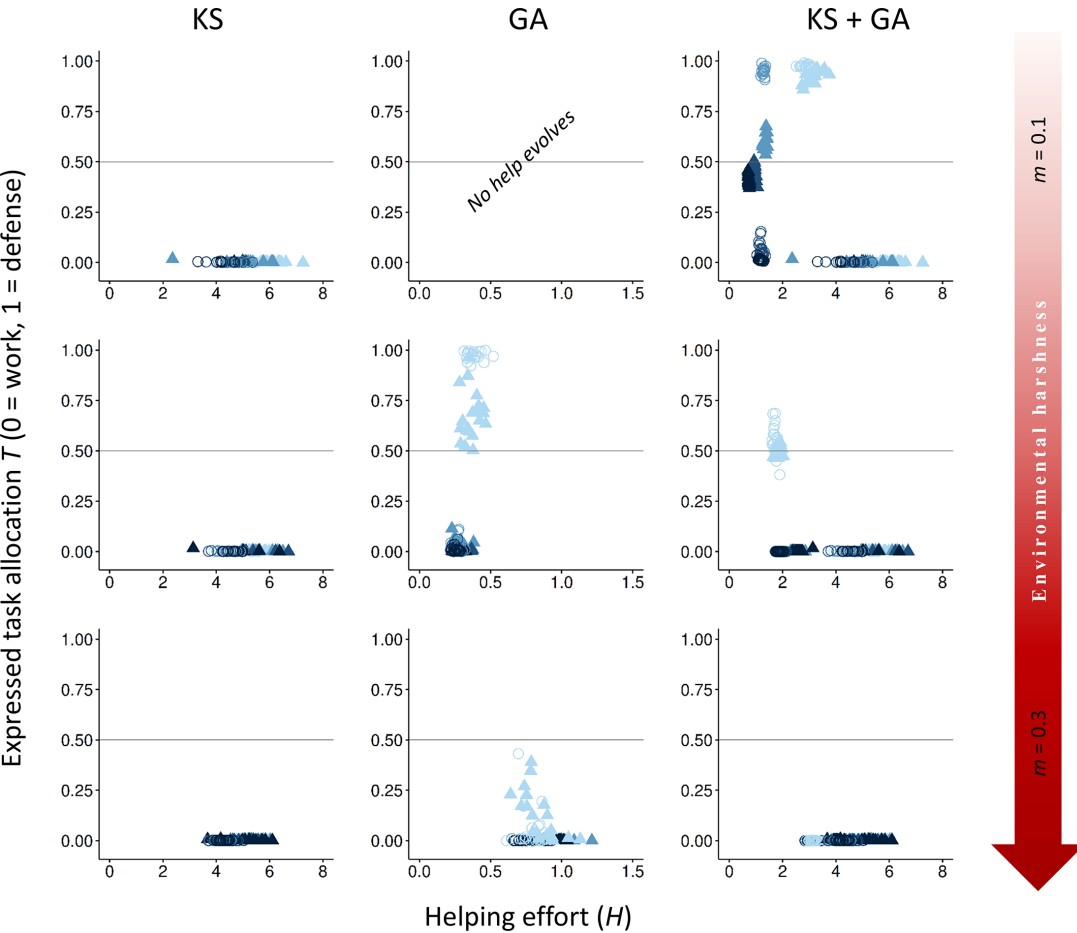

**Appendix 4—figure 1.** Effect of decreasing the benefits of group augmentation. The evolutionary equilibria for phenotypic levels of helping and task specialization are shown at three different levels of environmental quality, ranging from benign ($m = 0.1$) to harsh ($m = 0.3$), and for three different levels of cost of help on survival (lightest blue, $x_h = 1$, blue, $x_h = 3$; dark blue, $x_h = 5$; and darkest blue, $x_h = 7$), across 20 replicas. The vertical axis expresses the probability of individuals choosing a defensive task with a cost to survival versus a work task with a cost to their dominance rank. The optimum breeder productivity per unit of help provided was either when both tasks were performed to a similar extent, potentially selecting for division of labor (▲) or when no restrictions were introduced to the task performed by the group members (○). In each environment, additional details are given on the selective forces that play a role in the evolution of help and task specialization: help can only evolve by kin selection (KS), group augmentation (GA), or both (KS + GA). Input parameters are the same as in *Figure 2*, except for fixed value of $x_n = 1$ for the GA and KS + GA implementations. Additional details are provided in *Appendix 4—table 1*.

**Appendix 4—table 1.** Supplementary data for the effect of decreasing the benefits of group augmentation shown in **Appendix 4—figure 1**.

Mean values are shown for dispersal propensity, survival probability, group size (± SD), number of floaters (± SD), ratio between helpers' and floaters' dominance value (± SD), and within-group relatedness for three environmental qualities ranging from benign ($m = 0.1$) to harsh ($m = 0.3$) across 20 replicas. Selective forces at play include kin selection (KS), group augmentation (GA), or both (KS + GA). The optimum breeder productivity per unit of help provided was either when both tasks were performed to a similar extent, potentially selecting for division of labor (DoL), or when no restrictions were introduced to the task performed by the group members (No DoL).

| | KS | | | | | | GA | | | | | | KS + GA | | | | | |
|---|---|---|---|---|---|---|---|---|---|---|---|---|---|---|---|---|---|---|
| | $m = 0.1$ | | $m = 0.2$ | | $m = 0.3$ | | $m = 0.1$ | | $m = 0.2$ | | $m = 0.3$ | | $m = 0.1$ | | $m = 0.2$ | | $m = 0.3$ | |
| | DoL | No DoL | DoL | No DoL | DoL | No DoL | DoL | No DoL | DoL | No DoL | DoL | No DoL | DoL | No DoL | DoL | No DoL | DoL | No DoL |
| Dispersal | 0.93 | 0.90 | 0.92 | 0.89 | 0.90 | 0.84 | 0.01 | 0.01 | 0.22 | 0.15 | 0.62 | 0.46 | 0.07 | 0.07 | 0.22 | 0.22 | 0.53 | 0.50 |
| Survival | 0.74 | 0.74 | 0.65 | 0.65 | 0.57 | 0.57 | 0.90 | 0.90 | 0.77 | 0.78 | 0.63 | 0.65 | 0.89 | 0.89 | 0.77 | 0.77 | 0.64 | 0.64 |
| Group size | 1.24±0.20 | 1.43±0.30 | 1.18±0.13 | 1.29±0.18 | 1.16±0.07 | 1.30±0.09 | 9.83±0.06 | 9.81±0.06 | 5.25±1.52 | 6.23±1.09 | 2.13±0.90 | 2.91±0.93 | 16.18±0.22 | 16.38±0.13 | 6.51±0.08 | 6.66±0.06 | 2.64±0.05 | 2.89±0.06 |
| Number of floaters | 14962±1673 | 16240±2289 | 9998±731 | 10620±1049 | 7015±311 | 7518±356 | 303±25 | 301±24 | 4461±1742 | 3843±1109 | 6590±627 | 6255±545 | 5791±180 | 5915±121 | 7758±185 | 7880±181 | 9138±133 | 9361±169 |
| Rank ratio helpers vs floaters | 0.89±0.22 | 0.92±0.18 | 0.82±0.21 | 0.86±0.22 | 0.74±0.24 | 0.75±0.11 | 1.05±0.05 | 1.05±0.06 | 1.05±0.02 | 1.05±0.02 | 1.00±0.02 | 1.00±0.03 | 1.02±0.01 | 1.01±0.02 | 0.96±0.02 | 0.97±0.02 | 0.82±0.01 | 0.87±0.01 |
| Relatedness | 0.37 | 0.36 | 0.53 | 0.52 | 0.73 | 0.71 | 0 | 0 | 0 | 0 | 0 | 0 | 0.57 | 0.56 | 0.57 | 0.57 | 0.61 | 0.61 |

