## [Editor Report · eLife Assessment]

This **useful** study develops an individual-based model to investigate the evolution of division of labor in vertebrates, comparing the contributions of group augmentation and kin selection. The findings are **solid** in showing that, within the specific structure of the model and the parameter space explored, group augmentation can robustly favor the evolution of differentiated helper roles, particularly when age-dependent task switching and dominance dynamics are allowed to evolve. However, the evidence only partially supports the authors' broader claim that group augmentation is the primary driver of vertebrate division of labor. Several modelling assumptions, including the limited scope for synergistic task benefits, the restriction of helper effects to group-size-mediated benefits, and the relatively narrow exploration of cost and benefit parameters, constrain the potential for kin selection to generate division of labor and limit the generality of the conclusions.

---

## [Referee Report · Reviewer #2 (Public review)]

Summary:

This paper formulates an individual-based model to understand the evolution of division of labor in vertebrates. The model considers a population subdivided in groups, each group has a single asexually-reproducing breeder, other group members (subordinates) can perform two types of tasks called "work" or "defense", individuals have different ages, individuals can disperse between groups, each individual has a dominance rank that increases with age, and upon death of the breeder a new breeder is chosen among group members depending on their dominance. "Workers" pay a reproduction cost by having their dominance decreased, and "defenders" pay a survival cost. Every group member receives a survival benefit with increasing group size. There are 6 genetic traits, each controlled by a single locus, that control propensities to help and disperse, and how task choice and dispersal relate to dominance. To study the effect of group augmentation without kin selection, the authors cross-foster individuals to eliminate relatedness. The paper allows for the evolution of the 6 genetic traits under some different parameter values to study the conditions under which division of labour evolves, defined as the occurrence of different subordinates performing "work" and "defense" tasks. The authors envision the model as one of vertebrate division of labor.

The main conclusion of the paper is that group augmentation is the primary factor causing the evolution of vertebrate division of labor, rather than kin selection. This conclusion is drawn because, for the parameter values considered, when the benefit of group augmentation is set to zero, no division of labor evolves and all subordinates perform "work" tasks but no "defense" tasks.

Strengths:

The model incorporates various biologically realistic details, including the possibility to evolve age polytheism where individuals switch from "work" to "defence" tasks as they age or vice versa, as well as the possibility of comparing the action of group augmentation alone with that of kin selection alone.

Weaknesses from the previous round of review::

The model and its analysis are limited, which in my view makes the results insufficient to reach the main conclusion that group augmentation and not kin selection is the primary cause of the evolution of vertebrate division of labour. There are several reasons.

First, although the main claim that group augmentation drives the evolution of division of labour in vertebrates, the model is rather conceptual in that it doesn't use quantitative empirical data that applies to all/most vertebrates and vertebrates only. So, I think the approach has a conceptual reach rather than being able to achieve such conclusion about a real taxon.

Second, I think that the model strongly restricts the possibility that kin selection is relevant. The two tasks considered essentially differ only by whether they are costly for reproduction or survival. "Work" tasks are those costly for reproduction and "defense" tasks are those costly for survival. The two tasks provide the same benefits for reproduction (eqs. 4, 5) and survival (through group augmentation, eq. 3.1). So, whether one, the other, or both helper types evolve presumably only depends on which task is less costly, not really on which benefits it provides. As the two tasks give the same benefits, there is no possibility that the two tasks act synergistically, where performing one task increases a benefit (e.g., increasing someone's survival) that is going to be compounded by someone else performing the other task (e.g., increasing that someone's reproduction). So, there is very little scope for kin selection to cause the evolution of labour in this model. Note synergy between tasks is not something unusual in division of labour models, but is in fact a basic element in them, so excluding it from the start in the model and then making general claims about division of labour is unwarranted. In their reply, the authors point out that they only consider fertility benefits as this, according to them, is what happens in cooperative breeders with alloparental care; however, alloparental care entails that workers can increase other's survival *without group augmentation*, such as via workers feeding young or defenders reducing predator-caused mortality, as a mentioned in my previous review but these potentially kin-selected benefits are not allowed here.

Third, the parameter space is understandably little explored. This is necessarily an issue when trying to make general claims from an individual-based model where only a very narrow parameter region of a necessarily particular model can be feasibly explored. As in this model the two tasks ultimately only differ by their costs, the parameter values specifying their costs should be varied to determine their effects. In the main results, the model sets a very low survival cost for work (yh=0.1) and a very high survival cost for defense (xh=3), the latter of which can be compensated by the benefit of group augmentation (xn=3). Some limited variation of xh and xn is explored, always for very high values, effectively making defense unevolvable except if there is group augmentation. In this revision, additional runs have been included varying yh and keeping xh and xn constant (Fig. S6), so without addressing my comment as xn remains very high. Consequently, the main conclusion that "division of labor" needs group augmentation seems essentially enforced by the limited parameter exploration, in addition to the second reason above.

Fourth, my view is that what is called "division of labor" here is an overinterpretation. When the two helper types evolve, what exists in the model is some individuals that do reproduction-costly tasks (so-called "work") and survival-costly tasks (so-called "defense"). However, there are really no two tasks that are being completed, in the sense that completing both tasks (e.g., work and defense) is not necessary to achieve a goal (e.g., reproduction). In this model there is only one task (reproduction, equation 4,5) to which both helper types contribute equally and so one task doesn't need to be completed if completing the other task compensates for it; instead, it seems more fitting to say that there are two types of helpers, one that pays a fertility cost and another one a survival cost, for doing the same task. So, this model does not actually consider division of labor but the evolution of different helper types where both helper types are just as good at doing the single task but perhaps do it differently and so pay different types of costs. In this revision, the authors introduced a modified model where "work" and "defense" must be performed to a similar extent. Although I appreciate their effort, this model modification is rather unnatural and forces the evolution of different helper types if any help is to evolve.

I should end by saying that these comments don't aim to discourage the authors, who have worked hard to put together a worthwhile model and have patiently attended to my reviews. My hope is that these comments can be helpful to build upon what has been done to address the question posed.

[Editors' note: the authors have provided responses to the each of these points.]

---

## [Author Response]

The following is the authors’ response to the previous reviews

**Public Reviews:**

**Reviewer #2 (Public review):**
Summary:This paper formulates an individual-based model to understand the evolution of division of labor in vertebrates. The model considers a population subdivided in groups, each group has a single asexually-reproducing breeder, other group members (subordinates) can perform two types of tasks called "work" or "defense", individuals have different ages, individuals can disperse between groups, each individual has a dominance rank that increases with age, and upon death of the breeder a new breeder is chosen among group members depending on their dominance. "Workers" pay a reproduction cost by having their dominance decreased, and "defenders" pay a survival cost. Every group member receives a survival benefit with increasing group size. There are 6 genetic traits, each controlled by a single locus, that control propensities to help and disperse, and how task choice and dispersal relate to dominance. To study the effect of group augmentation without kin selection, the authors cross-foster individuals to eliminate relatedness. The paper allows for the evolution of the 6 genetic traits under some different parameter values to study the conditions under which division of labor evolves, defined as the occurrence of different subordinates performing "work" and "defense" tasks. The authors envision the model as one of vertebrate division of labor.The main conclusion of the paper is that group augmentation is the primary factor causing the evolution of vertebrate division of labor, rather than kin selection. This conclusion is drawn because, for the parameter values considered, when the benefit of group augmentation is set to zero, no division of labor evolves and all subordinates perform "work" tasks but no "defense" tasks.Strengths:The model incorporates various biologically realistic details, including the possibility to evolve age polytheism where individuals switch from "work" to "defense" tasks as they age or vice versa, as well as the possibility of comparing the action of group augmentation alone with that of kin selection alone.Weaknesses:The model and its analysis are limited, which in my view makes the results insufficient to reach the main conclusion that group augmentation and not kin selection is the primary cause of the evolution of vertebrate division of labor. There are several reasons.(1) First, although the main claim that group augmentation drives the evolution of division of labor in vertebrates, the model is rather conceptual in that it doesn't use quantitative empirical data that applies to all/most vertebrates and vertebrates only. So, I think the approach has a conceptual reach rather than being able to achieve such a conclusion about a real taxon.

We appreciate the reviewer’s point that our model does not incorporate quantitative empirical data across vertebrate taxa. This is indeed a limitation and reflects the current lack of fine-scale datasets on task division, the influence of life-history traits, and the fitness consequences of different cooperative activities in vertebrates. One of our aims, however, is precisely to stimulate such empirical work by highlighting the value of examining division of labor in species inhabiting harsh environments, considering age/size/dominance structure when evaluating variation in cooperative activities, and incorporating defense behaviors more consistently into analyses of helping, especially since defenders are often overlooked relative to the classic helpers-at-the-nest that provision offspring. The model therefore remains directly relevant to vertebrate systems because it departs from insect-inspired approaches that focus on fitness outcomes based solely in maximizing colony productivity. Instead, it incorporates direct fitness benefits to group members, an essential feature of vertebrate cooperative breeding and of other systems with fertile “workers,” as we clarified in the discussion.

(2) Second, I think that the model strongly restricts the possibility that kin selection is relevant. The two tasks considered essentially differ only by whether they are costly for reproduction or survival. "Work" tasks are those costly for reproduction and "defense" tasks are those costly for survival. The two tasks provide the same benefits for reproduction (eqs. 4, 5) and survival (through group augmentation, eq. 3.1). So, whether one, the other, or both helper types evolve presumably only depends on which task is less costly, not really on which benefits it provides. As the two tasks give the same benefits, there is no possibility that the two tasks act synergistically, where performing one task increases a benefit (e.g., increasing someone's survival) that is going to be compounded by someone else performing the other task (e.g., increasing that someone's reproduction). So, there is very little scope for kin selection to cause the evolution of labor in this model. Note synergy between tasks is not something unusual in division of labor models, but is in fact a basic element in them, so excluding it from the start in the model and then making general claims about division of labor is unwarranted. In their reply, the authors point out that they only consider fertility benefits as this, according to them, is what happens in cooperative breeders with alloparental care; however, alloparental care entails that workers can increase other's survival *without group augmentation*, such as via workers feeding young or defenders reducing predator-caused mortality, as a mentioned in my previous review but these potentially kin-selected benefits are not allowed here.

We understand the reviewer’s concern that our model restricts the scope for kin-selected benefits by not including task-specific synergy effects—specifically, help that directly increases the survival of group members (e.g., load-lightening via feeding young, or predator defense that reduces mortality of breeders or offspring independently of group augmentation). We agree that such effects can occur in some cooperative breeders, and that they can, in principle, generate indirect fitness benefits. However, even when helpers increase the survival of breeders or reduce parental investment per offspring, these effects generally translate into higher breeder productivity—either via increased fecundity, increased survival to the next breeding attempt, or increased investment in subsequent broods. Thus, although we treat benefits in terms of enhanced breeder productivity, this formulation implicitly captures a range of help-related effects that ultimately improve the reproductive output of the breeders, including those mediated through increased survival. For this reason, we believe that the model remains relevant for vertebrate systems despite not representing each pathway separately.

(3) Third, the parameter space is understandably little explored. This is necessarily an issue when trying to make general claims from an individual-based model where only a very narrow parameter region of a necessarily particular model can be feasibly explored. As in this model the two tasks ultimately only differ by their costs, the parameter values specifying their costs should be varied to determine their effects. In the main results, the model sets a very low survival cost for work (yh=0.1) and a very high survival cost for defense (xh=3), the latter of which can be compensated by the benefit of group augmentation (xn=3). Some limited variation of xh and xn is explored, always for very high values, effectively making defense unevolvable except if there is group augmentation. In this revision, additional runs have been included varying yh and keeping xh and xn constant (Fig. S6), so without addressing my comment as xn remains very high. Consequently, the main conclusion that "division of labor" needs group augmentation seems essentially enforced by the limited parameter exploration, in addition to the second reason above.

As we have explained in previous revisions, the costs associated with work and defense are not directly comparable because they affect different fitness components: work costs reduce dominance, whereas defense costs reduce survival. Whether a particular cost is “high” or “low” can only be evaluated by examining the evolved reaction norms and identifying the ranges over which these norms change. For this reason, we focused on parameter ranges that actually generate shifts in reaction norms rather than presenting large regions of parameter space where nothing changes.

We also reiterate that we did in fact explore broader parameter ranges than those shown in the main text. Additional analyses, including those specifically designed to identify conditions under which division of labor evolves under kin selection alone, are provided in the Supplementary Material. Specifically, Figure S1 addresses the point raised by the “need” of group augmentation benefits for defense to evolve, by increasing the baseline survival *x0*.

We now include one additional figure in the Supplementary Material with a lower value for the benefit of group size (*xn* = 1 instead of *xn* = 3), and we extended the range of *xh* to include lower values (*xh* = 1). As we can see in Figure S7 and Table S8, group augmentation benefits are still the primary reason for individuals to group (see dispersal values). For low benefits of group augmentation, defense evolves in harsh environments in the absence of kin selection, and in benign environments when both direct and indirect fitness benefits take place. We have also now expanded the results section to include these last results. Note that we also checked even lower values for *xh* under the only kin selection implementation, with results being qualitatively similar, but chose not to include them in the manuscript since it is already a very long Supplementary Material. Here are the averages for two examples with *xh* = 0.1 and when we promote division of labor:

**Author response table 1. sa2table1:** 

	task	help	dispersal	survival	relatedness
m=0.1	0.01	6.19	0.88	0.74	0.37
m=0.3	0.04	5.18	0.87	0.57	0.71

In short, the conclusion that division of labor requires group augmentation is not an artifact of limited parameter exploration. It arises because kin selection alone favors division of labor only under highly restrictive parameter combinations, whereas including direct fitness benefits substantially expands the conditions under which division of labor evolves. This pattern is consistent across the full set of parameter combinations we examined.

(4) Fourth, my view is that what is called "division of labor" here is an overinterpretation. When the two helper types evolve, what exists in the model is some individuals that do reproduction-costly tasks (so-called "work") and survival-costly tasks (so-called "defense"). However, there are really no two tasks that are being completed, in the sense that completing both tasks (e.g., work and defense) is not necessary to achieve a goal (e.g., reproduction). In this model there is only one task (reproduction, equation 4,5) to which both helper types contribute equally and so one task doesn't need to be completed if completing the other task compensates for it; instead, it seems more fitting to say that there are two types of helpers, one that pays a fertility cost and another one a survival cost, for doing the same task. So, this model does not actually consider division of labor but the evolution of different helper types where both helper types are just as good at doing the single task but perhaps do it differently and so pay different types of costs. In this revision, the authors introduced a modified model where "work" and "defense" must be performed to a similar extent. Although I appreciate their effort, this model modification is rather unnatural and forces the evolution of different helper types if any help is to evolve.

In previous models of division of labor in eusocial insects, the implicit benefit is also colony-level productivity (see Beshers & Fewell, 2001, for a review of division of labor in insects). Even in humans, division of labor functions as a means to increase efficiency toward achieving a shared goal. Our model adopts this same interpretation, as outlined in the Introduction, but extends it by considering that different tasks may impose different fitness costs, an aspect that has been largely overlooked in the existing literature. It is precisely because fitness outcomes are not fully shared among group members in vertebrates that distinguishing these cost structures matters. Unlike eusocial insects with sterile workers, vertebrate helpers can obtain direct fitness benefits, and the model explicitly accounts for these direct benefits—something absent from most insect-inspired approaches even when direct fitness benefits can also arise in some of those systems. Thus, our framework is not simply evolving “two types of helpers doing the same task,” but instead evolving specialization in different cooperative roles that carry different fitness consequences. It is therefore suitable for our model to treat contributions to breeder productivity as a common currency, while allowing individuals to specialize in different cost-distinct forms of help.

Finally, regarding synergy: with the extension introduced in the previous revision, we now incorporate the requirement that multiple forms of help must be performed for the group to achieve maximal reproductive output. This directly addressed the reviewer’s concern about synergistic dependencies between tasks and aligns our framework with the kinds of complementarity highlighted in other models of division of labor.

In summary, the structure of the model is consistent with both the theoretical literature on division of labor and the biological realities of vertebrate cooperative systems. We believe it is important for future models to explicitly consider the different fitness benefits and costs associated with distinct cooperative behaviors, and hope that our framework encourages more targeted empirical research on division of labor in vertebrates (e.g. inclusion of data on defense, life-history traits and environmental challenges) to better inform future modelling efforts.

I should end by saying that these comments don't aim to discourage the authors, who have worked hard to put together a worthwhile model and have patiently attended to my reviews. My hope is that these comments can be helpful to build upon what has been done to address the question posed.

We appreciate the reviewer’s thoughtful and constructive comments, as well as the time invested in evaluating our work. These insights have greatly helped us improve the clarity and overall quality of the manuscript. We hope that the revisions and additional clarifications we have provided adequately address all remaining concerns.